# Large-Batch, Iteration-Efficient Neural Bayesian Design Optimization

## Abstract

Bayesian optimization (BO) provides a powerful framework for optimizing black-box, expensive-to-evaluate functions. It is therefore an attractive tool for engineering design problems, typically involving multiple objectives. Thanks to the rapid advances in fabrication and measurement methods as well as parallel computing infrastructure, querying many design problems can be heavily parallelized. This class of problems challenges BO with an unprecedented setup where it has to deal with very large batches, shifting its focus from sample efficiency to iteration efficiency. We present a novel Bayesian optimization framework specifically tailored to address these limitations. Our key contribution is a highly scalable, sample-based acquisition function that performs a non-dominated sorting of not only the objectives but also their associated uncertainty. We show that our acquisition function in combination with different Bayesian neural network surrogates is effective in data-intensive environments with a minimal number of iterations. We demonstrate the superiority of our method by comparing it with state-of-the-art multi-objective optimizations. We perform our evaluation on two real-world problems - airfoil design and 3D printing - showcasing the applicability and efficiency of our approach. Our code is available at: https://github.com/an-on-ym-ous/lbn_mobo

## 1 Introduction

Design of objects and materials that lead to a specific *performance*, typically defined by multiple objectives, is a long-standing, critical problem in engineering Arnold (2018). For real-world design problems, the forward mechanisms that govern the design processes are either sophisticated physics-based simulations or time- and labor-intensive lab experiments. We call these underlying mechanisms *native* forward processes (NFP), which unlike *surrogate* models, are the most faithful design evaluation tools at our disposal. A powerful paradigm of design optimization is Bayesian optimization Jones et al. (1998) featuring a surrogate model that queries the NFP iteratively using a single data sample or a small batch of data. The choice of the next-iteration data is through optimizing a so-called *acquisition* function.

While Bayesian optimization literature focuses on solutions with a minimum number of NFP evaluations, a practically common but heavily underrepresented class of problems, especially in design optimization, are those in which heavy parallelization is possible but performing iteration is very demanding. A lower number of iterations is particularly beneficial in experimental scenarios where conducting lab experiments can be costly and time-consuming, making it desirable to minimize the number of lab visits. Thanks to emerging high throughput experimentation MacLeod et al. (2022), many of these problems lend themselves well to large-batch settings where it is feasible to produce a large batch of samples in an iteration with almost equivalent cost of generating a single sample. In such setups, it is typically desirable to increase the batch size as large as possible. This type of setup is abundant in real-world applications, such as materials science Raccuglia et al. (2016), drug discovery Dahl et al. (2014), robotics Marco et al. (2016), aerospace engineering Chen and Ahmed (2021), manufacturing Cucerca et al. (2020); Panetta et al. (2022), computational fluid dynamics Jofre and Doostan (2022); Du et al. (2021); Sun et al. (2023), etc. Despite the numerous real-world experiments that could gain from large-batch optimization, there is a remarkable scarcity of Bayesian optimization algorithms adept at managing large batches particularly for multi-objective optimization. To solve this class of problems effectively, we need to work toward shifting the paradigm from sample

efficiency to iteration efficiency. Existing methods have limitations in retrieving good solutions either in a few iterations, or handling very large training data, or dealing with multiple objectives.

We address these shortcomings by proposing a large-batch, neural multi-objective Bayesian optimization method (**LBN-MOBO**). The larger batch size results in fewer iterations needed to identify the Pareto front, enhancing the efficiency significantly. Similar to any BO framework, our method has two key components. First, demonstrating the insufficiency of current acquisition functions in dealing with large batch regimes, we propose a highly practical acquisition function based on multi-objective sorting of samples where not only the performance objective but also its associated uncertainty is considered when ranking the samples. By bringing in the uncertainty as an additional objective, LBN-MOBO can explore previously unseen regions, preventing it from getting trapped in local minima. Second, a surrogate model capable of handling very large batches of data and also computing predictive uncertainty. We illustrate that, while our pipeline is compatible with all the introduced Bayesian neural networks (BNNs), Deep Ensembles Lakshminarayanan et al. (2016) ultimately presents the most balanced trade-off between performance and scalability. We evaluate a range of state-of-the-art acquisition functions and surrogate models to show how current Bayesian optimizers struggle to exploit such large batch sizes. While we focus on evaluating *neural* BO frameworks in the paper, suitable for learning via large batches, we provide extensive evaluation of a set of promising standard BO methods (relying on Gaussian process surrogates) in the Appendix. We investigate two real-world problems one requiring hands-on lab work with a 3D printer and the second one an expensive CFD fluid dynamic simulation and show the Pareto front can be obtained with an order of magnitude less iterations.

Our proposed method offers a set of important advantages. First, it can retrieve a dense *Pareto front*, capturing the trade-off between multiple objectives, at each iteration which results in convergence in as few iterations as possible. Second, it can handle very large training data, enabling it to be applied to a wide range of engineering problems including those arising from high-dimensional design spaces. Third, it is highly parallelizable, shifting the computational bottleneck from the optimization algorithm to the computational infrastructure or experimentation capacities used to evaluate the NFP. Our contributions include:

- A novel and scalable Bayesian optimization algorithm designed for the parallel, large-batch optimization of multi-objective problems, with a focus on iteration efficiency.
- A novel, practical acquisition function designed to effectively manage large batch multi-objective optimizations without inducing a computational bottleneck. Remarkably, this function is architected to be embarrassingly parallelizable, is gradient-free, offering the flexibility to be paired with any arbitrary surrogate models, thereby broadening its applicability and utility in diverse optimization scenarios.
- Extensive and critical evaluation of the state-of-the art BO acquisition functions and surrogate models in large batch regimes.

## 2 RELATED WORK: MULTI-OBJECTIVE NEURAL BAYESIAN OPTIMIZATION

Standard BO methods face two major bottleneck when given large batch sizes for multi-objective optimization. Initially the acquisition function cannot scale with the data and becomes extremely slow and later the Gaussian process surrogate faces great difficulty in fitting the large amount of data. Thus, in this section we focus on a variety of neural surrogate models for Bayesian optimization. We start by reviewing suitable acquisition functions capable of handling multiple objectives for at least a batch size of two. In Section 3, we show how all of them can fail in a large batch setup.

### 2.1 MULTI OBJECTIVE BATCH ACQUISITION FUNCTIONS

**Expected Hypervolume Improvement (EHVI)** serves as an acquisition function in multi-objective Bayesian optimization (MOBO). EHVI calculates the expected improvement in the hypervolume of the Pareto front. It has gained significant attention due to its capability to handle multi-objective optimization problems effectively. Emmerich et al. (2005) initially proposed the concept of hypervolume improvement, and several advancements have been made since then. **qEHVI** is a batch version of EHVI, designed to make decisions about querying multiple points in the design space simultaneously Daulton et al. (2020).

**Non-dominated EHVI (NEHVI)** is a variant of EHVI that focuses on the improvement of non-dominated points Daulton et al. (2021). The introduction of NEHVI was a step forward in dealing with issues related to the scalability of EHVI by reducing the complexity from exponential to polynomial with respect to the batch size. Moreover, NEHVI has demonstrated superior performance in addressing high-dimensional problems. Following the development of qEHVI, **qNEHVI** emerged as the batch variant of NEHVI.

**Pareto Efficient Global Optimization (ParEGO)** transforms a multi-objective problem into a series of single-objective problems through scalarization functions, combining the strengths of Efficient Global Optimization (EGO) in a multi-objective setting Knowles (2006). The batch version of ParEGO, qParEGO, facilitates parallel evaluations of multiple points, significantly reducing the time required to find optimal solutions in multi-objective optimization scenarios Daulton et al. (2020).

In Section 3 , we demonstrate the limitations of these acquisition functions while scaled to larger batch sizes. As we will see, several methods either fail to conclude the optimization process or encounter extreme inefficiencies for batch sizes exceeding 1000.

## 2.2 BAYESIAN NEURAL SURROGATE MODELS

There have been numerous attempts to substitute Gaussian Processes (GPs) with neural networks to improve surrogate's scalability Li et al. (2023), but in order to convey uncertainty information, derivation of Bayesian Neural Networks is necessary. For inferring the posterior in a Bayesian neural network with stochastic parameters, several methods are available:

**Hamiltonian Monte Carlo (HMC)** is a Markov Chain Monte Carlo (MCMC) method used for sampling from posterior distributions and has been recognized as a computational gold standard in Bayesian inference Neal et al. (2011). HMC leverages Hamiltonian dynamics to propose candidate states, reducing the correlation between consecutive samples and improving sampling efficiency.

**Stochastic Gradient HMC (SGMC)** is a variant of HMC that incorporates stochastic gradients to scale to large datasets by working with mini-batches Chen et al. (2014). SGHMC addresses the challenges of noise introduced by mini-batch gradients, making it a scalable and robust approach for approximate Bayesian inference.

**Deep Ensembles** trains multiple neural networks independently and aggregates their predictions to approximate the posterior predictive distribution Lakshminarayanan et al. (2016). This technique serves as a practical and effective heuristic for uncertainty estimation in BO.

**Monte Carlo Dropout (MC Dropout)** serves as a technique for approximating uncertainty in neural network models Gal and Ghahramani (2016). It involves performing dropout at inference time and running multiple forward passes (Monte Carlo simulations) through the network, each time with different dropped-out nodes. By averaging the results of these passes, MC Dropout provides a measure of uncertainty associated with the model's predictions.

Aside from methods for inferring the posterior of neural networks with stochastic parameters, several fundamentally different strategies exist for adapting a neural network as a Bayesian surrogate model:

**Infinite Width Bayesian Neural Networks (IBNNs)** are another class of neural surrogate models that can be seen as a bridge between the realm of neural networks and Gaussian processes. Research has shown that as a neural network's width approaches infinity, the distributions of the functions represented by the network converge to a Gaussian Process Neal (2012); Lee et al. (2017). This phenomenon implies that IBNNs can be seen as GPs with a specific neural network-derived covariance function. The properties of IBNNs make them a scalable alternative to traditional GPs, especially in high-dimensional spaces.

**Deep Kernel Learning (DKL)** combines deep neural networks and Gaussian Processes to model complex and high-dimensional data sets, harnessing the representational power of deep learning and the uncertainty quantification of GPs Wilson et al. (2016b;a); Ober et al. (2021). In DKL, a neural network acts as a feature extractor, transforming input data into a feature space where a GP models the relationships between the transformed inputs and the output at the last layer. This approach offers advantages such as non-linear feature learning, uncertainty quantification, and flexibility in model architecture.

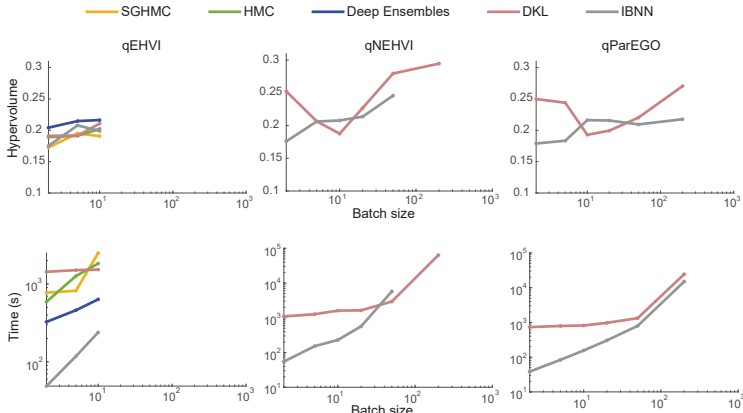

Figure 1: Optimizing 6D ZDT3 problem using a range of acquisition functions and surrogates.

In the following sections, we will illustrate how various combinations of these surrogate-acquisition pairs struggle to process even a moderate batch size of 500 samples, rendering them unsuitable for our class of problems which typically involves batch sizes an order of magnitude larger.

## 3 ANALYSIS OF CONTEMPORARY BATCH NEURAL MULTI-OBJECTIVE BO

In this section, we critically evaluate a selection of advanced acquisition functions adept at multi-objective batch optimization. Our findings highlight their limitations, specifically their inability to conduct optimization using large data batches. In combination with the acquisition functions, we evaluate a range of contemporary neural surrogates. The results indicate that the primary bottleneck in numerous scenarios is the acquisition function, followed by the surrogate. For this empirical validation, we rely on the ZDT3 problem. ZDT3 refers to one of the problems in the Zitzler-Deb-Thiele (ZDT) test suite Zitzler et al. (2000), widely used to evaluate and compare the performance of multi-objective optimizations. ZDT3 specifically consists of two objectives and a disjoint Pareto front (Section C.1 of the Appendix).

We start by computing the Pareto front of the 6-dimensional ZDT3 problem using qEHVI using several neural surrogates: DKL, HMC, IBNN, SGHMC, and Deep Ensembles (Section 2.2). For each batch size, the optimization is executed for 10 iterations, presenting the hypervolume of the best Pareto front achieved. Figure 1 (top left) demonstrates that when employing qEHVI as the acquisition function, the optimization stagnates at a batch size of 10. Notably, expanding to a batch size of 20 results in memory overflows independent of the surrogate. The compute time for this experiment is shown in Figure 1 (bottom left).

For applying qNEHVI and ParEGO, we rely on BoTorch implementation that supports only DKL and IBNN. Figure 1 reveals that both acquisition functions can handle batch sizes larger than qEHVI. Nonetheless, as depicted in Figure 1 (bottom row, middle and right), going beyond batch sizes larger than 200 increases the computational demand dramatically. In this work, each algorithm's GPU run-time was restricted to 44 hours for every batch size optimization process. More information regarding the hardware configuration is presented in Section D of the Appendix. Collectively, these experiments signify a gap in the capabilities of contemporary acquisition functions: They struggle with batch sizes approaching 500 samples, a scenario frequently encountered in our real-world applications, as elaborated in Section 5. Next, we introduce a novel acquisition function uniquely designed for working with extremely large batches.

## 4 METHOD: LARGE-BATCH NEURAL MOBO

Bayesian optimization for optimizing black-box NFP, $\Phi$, uses a surrogate model to create a prior over the objective function, which is updated with each new observation. An acquisition function $A_F$, derived from the surrogate model, guides the selection of the next samples, balancing exploration and

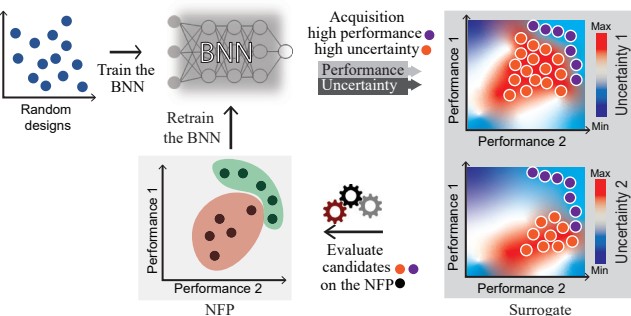

Figure 2: LBN-MOBO starts with training a Bayesian neural network ($f_{BNN}$) on random designs. We then run our acquisition function ($A_F$) and compute a $2MD$ Pareto front to explore promising (green) and under-represented regions (red) of the NFP. We then append the acquired candidates to the data set and retrain $f_{BNN}$. By incorporating uncertainty information alongside the Pareto front of the best performances (blue candidates), we identify promising candidates in areas of high uncertainty, where there is potential for additional information (red candidates).

exploitation. After evaluating the sample's performance, the surrogate model is updated. The process continues until a predetermined stopping criterion is reached.

Our method, LBN-MOBO, works on the same principles but is devised to achieve scalability. LBN-MOBO begins with a random sampling of the design space $\mathbf{U_S}(\mathcal{X})$ of the given NFP ($\Phi$). Subsequently, it fits an approximation of a Bayesian neural network surrogate $f_{BNN}$ to the randomly sampled dataset $\mathbf{X^0}$ in order to handle larger training data. Additionally, the Bayesian neural network $f_{BNN}$, and particularly its approximation through Deep Ensembles Lakshminarayanan et al. (2016), enables computing predictive uncertainties ($\mathbb{F}_\sigma(\mathbf{x})$) in a fully parallelized manner (Section 4.1). Upon training $f_{BNN}$, we utilize our acquisition function ($A_F$) to compute the candidates, which explores both promising and under-represented regions (Section 4.2). We append the calculated candidates to our data and utilize the updated dataset to train the BNN for the next generation.

Figure 2 illustrates the stages of the LBN-MOBO algorithm using two objectives as an example. Observe that some of the candidates may not be positioned on the Pareto front of the NFP (indicated by the red regions), but they are still retained in the dataset. This is because they contribute to enhancing the information of $f_{BNN}$ and decreasing its uncertainty level ($\mathbb{F}_\sigma(\mathbf{x})$). Algorithm 1 provides a concise summary of all the steps of LBN-MOBO.

---

**Algorithm 1** Large-batch, neural multi-objective Bayesian optimization (LBN-MOBO).

---

**Input**
$S$     // Batch size
$Q$     // Number of iterations of the main algorithm
$\mathcal{X}$    // $\mathcal{X} \in \mathcal{R}^n$, $n$ dimensional design space
$\Phi$    // Native Forward Process, e.g., a simulation
**Output** $P_S$, $P_F$ // Pareto set(designs) and Pareto front(performances) of NFP
**begin**
    $\mathbf{X^0} \leftarrow \mathbf{U_S}(\mathcal{X})$ // Draw $S$ random samples from the design space.
    $\mathbf{Y^0} \leftarrow \Phi(\mathbf{X^0})$ // Query $\Phi$ and form the data set.
    $dataset \leftarrow (\mathbf{X^0}, \mathbf{Y^0})$
    $f^0_{BNN} \overset{\text{train}}{\Longleftarrow} dataset$ // Train the BNN surrogate.
    **for** $i \leftarrow 1$ **to** $Q$ **do**
        $P^i_S \leftarrow A_F(f^{i-1}_{BNN}, S)$
        $P^i_F \leftarrow \Phi(P^i_S)$ // Calculate the performance on the NFP.
        $dataset \leftarrow (P^i_F, P^i_S)$ // Append new data to the old.
        $f^i_{BNN} \overset{\text{train}}{\Longleftarrow} dataset$ // Train the BNN surrogate.
    **end**
**end**

---

### 4.1 Bayesian neural network surrogate $f_{BNN}$

Given that Deep Ensembles Lakshminarayanan et al. (2016) presents the most balanced trade-off between performance and scalability (Section 5.1), it will serve as the primary acquisition function in our pipeline. Here, we delve into its implementation and make slight modifications to enhance its performance further. In this work, we employ a modified version of Deep Ensembles Lakshminarayanan et al. (2016) as an approximation of a BNN Snoek et al. (2015). Deep Ensembles consist of an ensemble of $K$ neural networks, $\hat{f}_k$, each capable of providing a prediction $\mu_k(\mathbf{x})$ and its associated *aleatoric* uncertainty $\sigma_k(\mathbf{x})$ in the form of a Gaussian distribution $\mathcal{N}(\mu_k(\mathbf{x}), \sigma_k(\mathbf{x}))$. In our case, in order to guide our optimizer to explore under-represented regions, we only require *epistemic* uncertainty. In the areas with higher epistemic uncertainty the networks in the ensemble fit differently due to a lack of information. Therefore, the regions with epistemic uncertainty are where we could potentially find better solutions. We don't include the aleatoric uncertainty because it finds the built-in noise in the NFP and cannot be reduced by taking more samples. In this work, we assume the NFPs do not feature significant noise.

Thus, we only train $K$ neural networks using the traditional mean squared error (MSE) loss.

$$\mathcal{L}_k^{MSE} := (\mathbf{y}^* - \mu_k(\mathbf{x}))^2. \tag{1}$$

Next, we extract the epistemic uncertainty $\mathbb{F}_\sigma(\mathbf{x})$ from the networks in the ensemble:

$$\mathbb{F}_\mu(\mathbf{x}) := \frac{1}{K} \sum_k \mu_k(\mathbf{x}), \tag{2a}$$

$$\mathbb{F}_\sigma(\mathbf{x}) = \frac{1}{K} \sum_k (\mu_k^2(\mathbf{x}) - \mathbb{F}_\mu^2(\mathbf{x})). \tag{2b}$$

Apart from this modification, we find that providing a diverse set of activation functions across $K$ members of the ensemble significantly helps with obtaining higher quality uncertainty. More details are provided in Section D of the appendix.

### 4.2 $2M$D acquisition function

An acquisition function should predict the worthiest candidates for the next iteration of the Bayesian optimization Shahriari et al. (2015). This translates to not only selecting designs with high performance on the surrogate model but also considering the uncertainty of the surrogate model. Candidates in uncertain regions of the surrogate model may contain appropriate solutions and a powerful acquisition function should be able to explore these regions effectively. Several popular acquisition functions such as Expected Improvement Jones et al. (1998) and Upper Confidence Bound Brochu et al. (2010) operate on this principle.

Without the loss of generality, we assume a problem that seeks to *maximize* performance objectives. Our acquisition function employs the widely-used NSGA-II Deb et al. (2002) and specifically its multi-objective non-dominated sorting method (Section A.1). This sorting is the key to find the Pareto front of the surrogate at a given iteration of LBN-MOBO. The main insight of our acquisition method is that instead of finding an $M$ dimensional Pareto front corresponding to $M$ objectives (each given by $\mathbb{F}_\mu^m(\mathbf{x})$, $m \in [1, M]$), it finds a $2M$ dimensional Pareto front where $M$ dimensions correspond to performance objectives and the other $M$ dimensions correspond to the uncertainty of those objectives (each given by $\mathbb{F}_\sigma^m(\mathbf{x})$, $m \in [1, M]$). In other words, our acquisition function $A_F$ *simultaneously* maximizes the predicted objectives and their associated uncertainties (both given by our surrogate $f_{BNN}$):

$$A_F := \arg\max_{\mathbf{x}} \left\{ \mathbb{F}_\mu^m(\mathbf{x}), \ \mathbb{F}_\sigma^m(\mathbf{x}) \right\}, m \in [1, M]. \tag{3}$$

Note that in practice NSGA-II experiences a sample size bottleneck and struggles to scale effectively as the population expands. To overcome this limitation, we propose to compute in parallel independent acquisitions (different NSGA-II seeds) with smaller batch sizes, and combine the results. Ultimately, similar to our surrogate model, our acquisition function ($A_F$) is fully parallelizable, and its performance

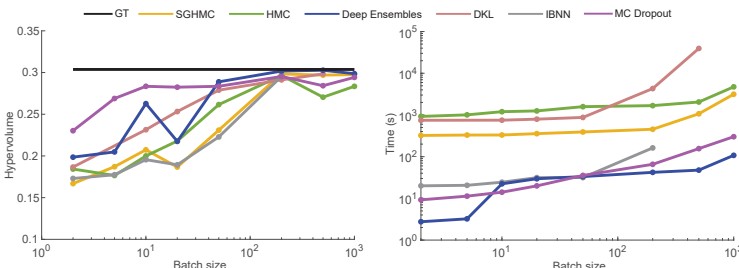

Figure 3: ZDT3 benchmark optimization via $2M$D acquisition function and various neural BO surrogate models. The fitting time of each model was recorded to assess computational efficiency.

remains unhampered even when batch size increases. Therefore, the sole limiting factor for executing LBN-MOBO is our parallel processing or experimentation capability when querying the NFP.

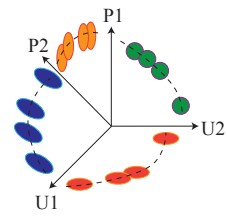

The inset figure provides an intuitive explanation by showing a schematic four-dimensional acquisition function. Clearly, we are interested in evaluating the orange samples (currently measured only using the surrogate) on the NFP as they are suggested by $A_F$ to be dominant in at least one *performance* dimension (P1 or P2). On the other hand, the blue, red, and green samples are chosen partially or entirely due to their high uncertainty in at least one dimension (U1 or U2). These samples correspond to unexplored regions in the design space. They are beneficial in two ways: either they prove to be part of the Pareto front once being evaluated on the NFP, or they contribute to filling the gap between the surrogate and the NFP, leading to a more informative surrogate model. This enhances the quality of the surrogate model, making it as similar as possible to the NFP, thereby improving its predictive power for subsequent iterations.

## 5 EVALUATION AND DISCUSSION

Before starting with two challenging real-world problems we evaluate the most scalable surrogate capable of handling large batch sizes in a reasonable time. While the focus of this section is on optimizing real-world problems using our neural Bayesian optimization, a comprehensive evaluation of various standard state-of-the-art Bayesian optimization techniques, as well as multi-objective evolutionary algorithms, can be found in Section C of the Appendix.

### 5.1 SELECTING THE MOST SUITABLE SURROGATE

A pivotal aspect, distinct from the acquisition function selection, is the choice of the neural surrogate. Even though our pipeline can work with any surrogate models (Appendix Section B.2), our objective is identifying the model that has the most efficiency and scalability. Analogous to Section 3, we assess the performance of various surrogate models, this time employing our $2M$D acquisition function. As depicted in Figure 3 (left), by using our proposed acquisition function, the bottleneck associated with the acquisition function is entirely alleviated, and all methods, with the exception of IBNN, have successfully completed the optimization for batch sizes up to 1000. Figure 3 (right) shows the optimization time for different surrogates, each undergoing a 10-iteration optimization across a spectrum of batch sizes. Deep Ensembles and MC dropout prove to be the most time-efficient models, adeptly conducting optimizations for batch sizes up to 1000 and finding the best Pareto front in the least number of iterations. These experiments substantiate Deep Ensembles and MC dropout as the clear choice to be paired with our novel surrogate model. A detailed explanation of how we use the MC dropout in combination with our $2M$D acquisition is provided in the Appendix Section B.1. We have provided more challenging experiments in the appendix with larger input and output dimensions (Appendix Section C.3).

### 5.2 REAL-WORLD EXPERIMENT SET UP

As discussed earlier, we found LBN-MOBO to be useful in two important classes of experiment: those involving cumbersome lab work or expensive simulations which can be parallelized.

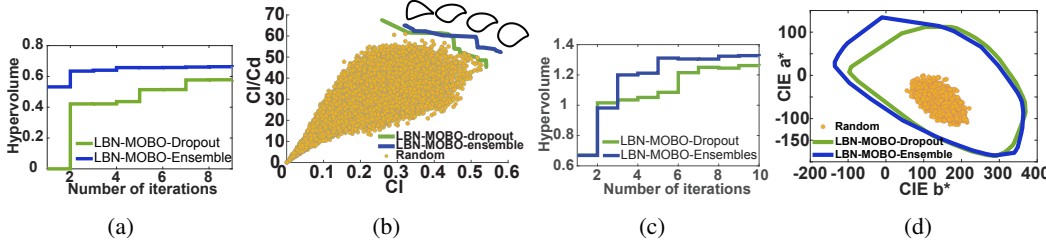

Figure 4: We present the evolution of hypervolume and the Pareto front for both the airfoil and the printer color gamut problems, utilizing LBN-MOBO with Deep Ensembles and MC dropout as the surrogate models. Figure (a) shows the hypervolume expansion of the Airfoil problem, and (b) represents the Pareto front calculated using each surrogate. Similarly, (c) depicts the hypervolume expansion of the printer color gamut problem, and (d) displays the gamut actually discovered by LBN-MOBO using both surrogates.

**3D printer's color gamut experiment** is an examples of type 1 experiments. The color gamut is the range of all colors that a device, such as a printer, can produce. The colorfulness is quantified using the CIE a*b* color space CIE (2004). In this space the diversity of colors directly translates into the *area* of the CIE a*b* plot. The Bayesian optimization iteratively enlarges this area as it discovers more saturated colors [1]. 3D printer is capable of generating many small (e.g., 1x1 mm) patches of color in a single operation making it a perfect show case for the benefits of large-batch optimization that LBN-MOBO can perform. In Sections 5.3 and C we show how this approach outperforms all other algorithms. More details of this experiment can be found in Section C.5.2 of Appendix.

**Airfoil's** lift ($C_l$) and drag ($C_d$) coefficient optimization falls into the second category of problems. In this setup we have control over the shape of the airfoils through 6 design parameters. The goal is to optimize the the designs to generate the largest $C_l$ and $C_l/C_d$ ratio. The evaluation of a single shape using a CFD simulator, i.e., NFP, is lengthy. Thus, to minimize the total optimization time we can rely on parallel computing to create large batches of data and find the best Pareto front in a few iterations. More details of the airfoil experiment can be found in Section C.5.1 of Appendix.

## 5.3 LBN-MOBO FOR REAL-WORLD PROBLEMS

In this study, we delve into the practical capabilities of LBN-MOBO by applying it to two complex real-world problems: airfoil and printer's color gamut, employing larger batch sizes of 15,000 and 20,000, respectively. The primary focus of this section is a comparison between LBN-MOBO with MC dropout and LBN-MOBO with Deep Ensembles. Both variations exhibit commendable performance in addressing the complexities of the selected problems, with Deep Ensembles showing a better performance. A comprehensive evaluation of additional methods on these problems is presented in Section C.6.1 andC.6.2 of Appendix. There we also establish the superiority of our method over a few other algorithms that can operate in this large batch regime.

Figure 4 illustrates the analysis for the airfoil problem, showcasing the performance of LBN-MOBO with MC dropout and with Deep Ensembles. This experiment introduces the complex problem of mapping the airfoil shapes to intricate aerodynamic properties. Both variations of LBN-MOBO runs with batch size equal 15,000. The candidate samples of each optimization iteration is simulated by the high-fidelity CFD solver OpenFOAM OpenFOAM Foundation (2021). Remarkably, as depicted in Figure 4a, both LBN-MOBO variations are capable of handling this huge batch of data and uncover superior Pareto fronts in a limited number of iterations. We can observe the superior performance of LBN-MOBO utilizing Deep Ensembles due to its higher-quality epistemic uncertainty.

For the printer's color gamut exploration, LBN-MOBO initiates with 10,000 samples, with each subsequent iteration managing a batch size of 20,000 samples. The high dimensionality of this design space (44) presents a formidable challenge, rendering it an intriguing test case. The performance

---

[1]For this problem, we solve four Bayesian optimizations for four quadrants in order to advance the Pareto front in four different segments.

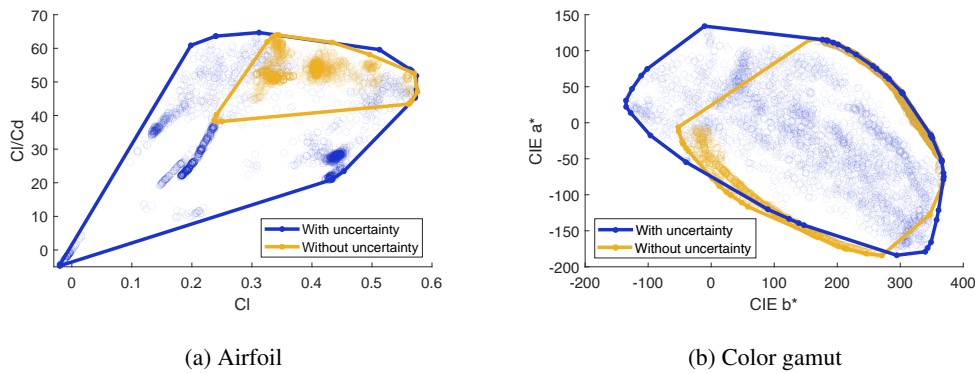

(a) Airfoil           (b) Color gamut

Figure 5: Ablation studies on the effect of uncertainty in our $2M$D acquisition function, using our real-world problems.

space for this experiment is represented by the 2-dimensional a*b* color space. Figure 4c vividly demonstrates the expedited increase in the hypervolume (are) of the color gamut achieved by LBN-MOBO. The final gamut estimation for both variations of LBN-MOBO after 10 iterations is illustrated in Figure 4d, revealing the effectiveness of our method in estimating a significantly large gamut is small number of iterations.

## 5.4 THE IMPACT OF UNCERTAINTY ON THE PERFORMANCE OF LBN-MOBO

One of the key factors enhancing the performance of LBN-MOBO is its use of uncertainty to effectively explore under-represented parts of the design space. We investigate the impact of uncertainty on the computation of the Pareto front for both airfoil design and color gamut exploration. Both experimental setups mirror the conditions described in Section 5.3, except that they exclude uncertainty information. The candidate distribution from iteration 4 to 8 is illustrated in Figures 5a and 5b. For a clearer depiction of the samples' spatial distribution, we have illustrated their convex hull. Note that in the absence of uncertainty, the candidates have a tendency to cluster within particular areas. This clustering leads to diminished diversity and, as a consequence, a reduction in the capacity for exploration (as represented by the yellow samples). Conversely, when uncertainty is incorporated into the process, we observe an increase in the diversity of the candidates and consequently, a broader Pareto front is discovered (represented by blue samples). Furthermore, uncertainty guides the candidates to progressively bridge the information gap in the surrogate models, making them increasingly similar to the NFP. This factor further enhances the quality of the Pareto front retrieved through the LBN-MOBO process.

We also observe that when uncertainty is excluded from the process, the budget for surrogate Pareto front optimization is concentrated solely on performance dimensions. This concentration may occasionally lead to a slight local enhancement in optimization, as illustrated in the bottom-left quarter of Figure 5b.

**Limitations.** LBN-MOBO emerges as a potent optimizer for problems where an increase in the batch size does not significantly inflate simulation or experimentation costs, but iterations are expensive. Notably, LBN-MOBO not only retrieves a superior Pareto front but also enhances the surrogate model throughout the optimization process, making it closely resemble the NFP. This implies that, within the context of active learning, this methodology could be implemented: starting with a random dataset and incrementally training the network with missing data until it converges to the NFP. Looking forward, there are a few key aspects of this method that warrant further exploration. First, the potential of LBN-MOBO in managing design constraints needs to be assessed. Second, an analysis of its performance in the presence of noisy data could be undertaken, and possibly, it could be extended to enhance its robustness against noise. Finally, while our current acquisition function is tuning-free, it is intriguing to explore explicit methods that manipulate the balance between exploration and exploitation and see how this balance affects the overall performance of LBN-MOBO.

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

APPENDIX

## A    COMPLEMENTARY RELATED WORKS

In addition to the neural BO models discussed in Section 2, several advanced BO algorithms, as well as stochastic multi-objective optimizations, are adept at handling multi-objective Bayesian optimization for moderate batch sizes. In this section we review the most relevant state of the art methods that can address our class of problems. In Sections C, we assess these methods using synthetic ZDT and DTLZ test suits, Printer color gamut, and Airfoil problems. We observe their stagnation with increasing batch sizes, generally lacking scalability for batches exceeding 1000 samples. Ultimately, we demonstrate that in terms of Pareto front optimality, these methods are not comparable to our LBN-MOBO algorithm.

### A.1    STOCHASTIC MULTI-OBJECTIVE OPTIMIZATIONS

**Non-dominated sorting genetic algorithm II (NSGA-II)** Deb et al. (2002) is an exceptionally popular method for multi-objective optimization. It belongs to multi-objective evolutionary algorithms (MOEA), which have been applied to a variety of problems, from engineering Schulz et al. (2018) to finance Subbu et al. (2005). Despite their widespread use, MOEA have certain limitations. One major challenge is the computational cost of MOEAs, as they typically require a large number of function evaluations, and iterations to converge to a good solution Konakovic Lukovic et al. (2020). This can make MOEAs impractical for problems with computationally expensive objective functions or high-dimensional design spaces. Moreover, they are prone to trap in local minima. This can be particularly problematic for problems with multiple local optima or non-convex objective functions.

### A.2    ADVANCED BAYESIAN MULTI-OBJECTIVE OPTIMIZATIONS

Bayesian optimization (BO) is adept at efficiently searching for the global optimum while minimizing the number of function evaluations Jones et al. (1998). However, extending BO to multi-objective batch optimization is not straightforward. **USeMO** Belakaria et al. (2020) is one of the state of the art extensions of BO that is capable of solving multi objective problems. It employs NSGA-II to identify the Pareto front on the surrogate and uses uncertainty information to select a subset of candidates for the next iteration. **TSEMO** Bradford et al. (2018) takes a different approach by using Thompson sampling and NSGA-II on Gaussian process (GP) surrogates to find the next batch of samples that maximize the hypervolume. However, these methods struggle to maintain diversity and fail to capture part of the final Pareto frontKonakovic Lukovic et al. (2020).

**Diversity-guided multi-objective Bayesian optimization (DGEMO)** seeks to address this issue by dividing the performance and design spaces into diverse regions and striving to identify candidates in as diverse locations as possible while maintaining the performance Konakovic Lukovic et al. (2020). However, its computational time grows exponentially with the increase in batch size.

**Thompson Sampling (TS)** has been the subject of significant research, with key contributions aiming to enhance the scalability of Bayesian Optimization. Hernández-Lobato et al. (2017) demonstrated its scalability in the chemical space through parallel and distributed computing, effectively handling large parallel measurements in BO. Deshwal et al. (2021) further extended its potentials in combinatorial BO settings by incorporating Mercer features, opening up new possibilities in molecular optimization. Vakili et al. (2021) addressed scalability challenges by integrating sparse Gaussian process models with Thompson Sampling. They provide a theoretical and empirical analysis proving that this scalable TS shows much less computational complexity while maintaining its performance quality. Scalable TS introduces new possibilities especially in the combinatorial space of high-throughput molecular design problems.

### A.3    BAYESIAN OPTIMIZATION AND PARETO FRONT OF PREDICTION AND UNCERTAINTY

Gupta et al. (2018) proposed a unique algorithm that employs two distinct acquisition techniques to generate candidates for subsequent iterations. The primary insight of their method is to effectively handle problems characterized by a wide variety of local extrema, ranging from minimal to substantial in number. Their approach integrates the Gaussian Process Upper Confidence Bound (GP-UCB) with an additional Pareto front. This Pareto front is derived from optimizing predictions and uncertainties as separate objectives.

However, unlike the LBN-MOBO approach, Gupta et al.'s algorithm does not specifically focus on large batch optimization. Furthermore, it doesn't leverage Bayesian Neural Networks (BNNs) as surrogate models, which are key in enhancing scalability to the levels necessary for addressing complex real-world problems. Another distinct aspect of their work is that it appears primarily geared towards problems with single objectives, rather than the multi-objective scenarios that LBN-MOBO is designed to tackle.

## B COMPLEMENTARY METHODS

### B.1 MC DROPOUT

Incorporating MC dropout Gal and Ghahramani (2016) as a substitute for Deep Ensembles aims to leverage its inherent characteristics for assessing model uncertainty. MC dropout performs model uncertainty approximation by enabling dropout at the inference phase, generating diverse network predictions over multiple forward passes. These sub-networks will then ensemble in the same manner as our Deep Ensembles surrogate (Section 4.1) to calculate the mean and uncertainty of the predictions.

**MC Dropout as Surrogate Model** For seamless integration with LBN-MOBO, we employ a Neural Network with dropout layers, designated as $f_{MC}$. During the inference the dropout layers remain active, resulting in varied network structures for each forward pass. This stochasticity during inference results in a distribution of predictions for any given input, enabling the estimation of epistemic uncertainty.

A primary advantage of utilizing MC dropout lies in its capacity to compute uncertainties in parallel, akin to Deep Ensembles, hence maintaining the scalability of LBN-MOBO. After fitting $f_{MC}$ to the initial samples, the subsequent steps in LBN-MOBO remain consistent, with the updated surrogate model being utilized to compute the novel acquisition function.

**Epistemic Uncertainty through MC Dropout** MC dropout facilitates the calculation of epistemic uncertainty by observing the variance in predictions across multiple stochastic forward passes. Given a set of $T$ stochastic forward passes, the epistemic uncertainty for an input $\mathbf{x}$ can be computed as follows:

$$\mathbb{F}_{\mu_{MC}}(\mathbf{x}) := \frac{1}{T}\sum_t \mu_t(\mathbf{x}) \tag{4}$$

$$\mathbb{F}_{\sigma_{MC}}(\mathbf{x}) = \frac{1}{T}\sum_t (\mu_t^2(\mathbf{x}) - \mathbb{F}_{\mu_{MC}}^2(\mathbf{x})). \tag{5}$$

**Modified Acquisition Function with MC Dropout** With MC dropout incorporated as the surrogate model, the novel acquisition function now utilizes the uncertainties and predictions obtained from $f_{MC}$. The modified acquisition function aims to balance the trade-off between exploiting regions of high predicted performance and exploring regions with high uncertainty, as determined by the MC dropout model.

$$A_{\mathrm{F}MC} := \arg\max_{\mathbf{x}} \left\{ \mathbb{F}_{\mu_{MC}}^m(\mathbf{x}), \mathbb{F}_{\sigma_{MC}}^m(\mathbf{x}) \right\}, m \in [1, M]. \tag{6}$$

This integration of MC dropout with the novel acquisition function ensures that the benefits of epistemic uncertainty estimation are harnessed effectively while maintaining the scalability and parallelism inherent to the LBN-MOBO framework.

### B.2 COMBINATION OF OTHER BNNS WITH $2M$D ACQUISITION FUNCTION

One of the main advantages of $2M$D acquisition function is that it only queries a surrogate models that provides the prediction and uncertainty (without the need of gradient information). Figure 6

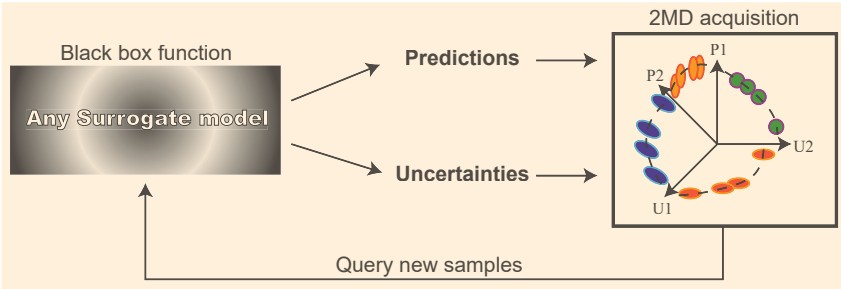

Figure 6: $2MD$ acquisition function can be paired with any surrogate model.

demonstrates how 2$M$D performs by iteratively using these queried information and evolves the solution until a practically good convergence. This property of 2MD acquisition allows it to be paired easily with any form of surrogate model.

## C  COMPLEMENTARY EVALUATION DETAILS

### C.1  ZDT PROBLEMS

The ZDT (Zitzler-Deb-Thiele) test suite Zitzler et al. (2000) is a set of benchmark problems commonly used for testing the performance of multi-objective optimization algorithms. The suite consists of six problems (ZDT1 through ZDT6), each having two objective functions. Here, we describe the formulations for ZDT1, ZDT2, and ZDT3.

#### C.1.1  ZDT1

ZDT1 is a 30-dimensional problem defined as follows:

Objective 1:

$$f_1(x) = x_1$$

Objective 2:

$$f_2(x) = g(x) \left[ 1 - \sqrt{\frac{x_1}{g(x)}} \right]$$

where:

$$g(x) = 1 + \frac{9}{n-1} \sum_{i=2}^{n} x_i$$

and $x_i$ is in the range [0,1].

#### C.1.2  ZDT2

ZDT2, like ZDT1, is also a 30-dimensional problem and is defined as follows:

Objective 1:

$$f_1(x) = x_1$$

Objective 2:

$$f_2(x) = g(x) \left[ 1 - \left( \frac{x_1}{g(x)} \right)^2 \right]$$

where:

$$g(x) = 1 + \frac{9}{n-1} \sum_{i=2}^{n} x_i$$

and $x_i$ is in the range [0,1].

### C.1.3 ZDT3

ZDT3, a 30-dimensional problem, introduces a discontinuous Pareto front. The objectives are defined as:

Objective 1:

$$f_1(x) = x_1$$

Objective 2:

$$f_2(x) = g(x) \left[ 1 - \sqrt{\frac{x_1}{g(x)}} - \frac{x_1}{g(x)} \sin(10\pi x_1) \right]$$

where:

$$g(x) = 1 + \frac{9}{n-1} \sum_{i=2}^{n} x_i$$

and $x_i$ is in the range [0,1].

### C.1.4 BENCHMARKING LBN-MOBO ON ZDT3

In this study, we highlight the superior performance of the LBN-MOBO compared to a set of state-of-the-art Multi-objective Bayesian optimizations, namely USeMO, DGEMO, TSEMO, and NSGA-II, on the ZDT3 test. We demonstrate how LBN-MOBO adeptly manages large design spaces while maintaining a significantly shorter optimization time compared to its counterparts. This investigation involves two ZDT3 problem configurations. The first experiment focuses on a 6-dimensional design space, while the second broadens this space to 30 dimensions, thereby increasing the complexity of the search space. We maintain a fair comparison by limiting the batch size to 1000 samples for all algorithms despite the fact that LBN-MOBO inherently possesses the ability to handle much larger batches. Using 1000 samples is an approximate limit of tractability for most of the competing methods. Moreover throughout this experiment we use an equal number of iterations for all methods, except in cases where a method becomes intractable due to unmanageable computational load.

Figure 7 (top) shows the superior performance of LBN-MOBO and DGEMO in contrast with the other algorithms for the 6-dimensional ZDT3 problem. In this figure, the illustrated Pareto fronts are the final results of 10 optimization iterations. When confronted with the 30 dimensional problem (bottom row), the optimization methods must navigate a significantly larger space within the same sampling constraints. For this problem, we have shown the results of optimizations at iteration 5.

In Figure 8 (left and middle), we further clarify the performance of different methods by showing the evolution of the hypervolume of the Pareto front at each iteration. For the 6D problem, LBN-MOBO manages to find the Pareto front after a single iteration. For the 30D problem, LBN-MOBO finds the Pareto front after two iterations and maintains its dominance over the other methods until the 6th iteration where DGEMO reaches it. We note that to approximate the hypervolume, we employed Simone (2023) that uses random sampling. As a result, occasional minor fluctuations may arise. It's worth mentioning that for the 30D problem we were unable to complete 10 iterations for UseMO, TSEMO, and DGEMO due to their exponential rise in computational time.

Figure 8 (right) depicts the run-time of all methods for 6D ZDT3 for 10 iterations. Even in this fairly straightforward problem, the computational time for UseMO, TSEMO, and particularly DGEMO surged dramatically. Conversely, the total computational time for NSGA2 was less than 14 seconds. The last and longest iteration of LBN-MOBO was 167s for training the ensemble models and 13.5s for acquisition.

### C.1.5 FURTHER EVALUATION USING ZDT1 AND ZDT2

In this section, we showcase ZDT1 and ZDT2, two problems from the ZDT test suite Zitzler et al. (2000). Both problems involve conflicting objectives, with the only distinction that ZDT1 has a convex Pareto front, whereas ZDT2 has a non-convex one. Both tests are conducted using their original 30-dimensional design space. The problem setup configuration is entirely identical to that of the ZDT3 problem Section 4.2 in the paper. As illustrated in Figure 9, similar to the case of ZDT3, LBN-MOBO demonstrates its superiority in both the ZDT1 and ZDT2 problems.

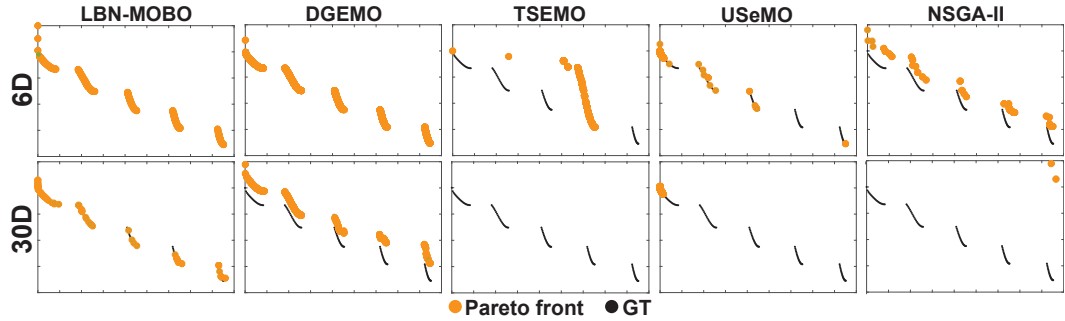

Figure 7: Obtained Pareto front by different methods on 6 dimensional and 30 dimensional ZDT3 problem. Batch size for all experiments is fixed at 1000.

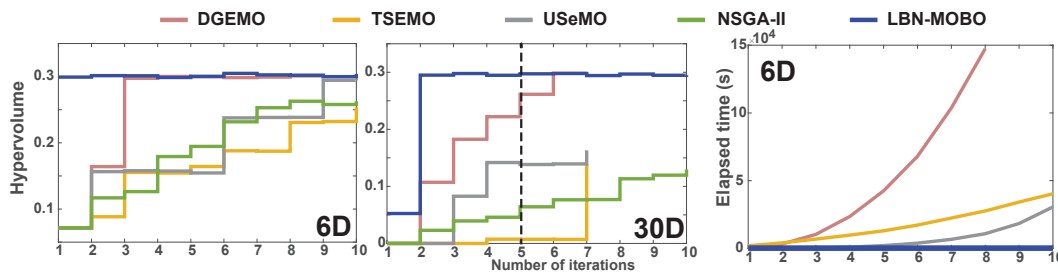

Figure 8: The left and middle plot represent the hypervolume of the 6 and 30 dimensional ZDT3 problem, respectively. The plot on the right shows the elapsed time for 6D problem for all methods (the NSGA-II plot is masked under LBN-MOBO).

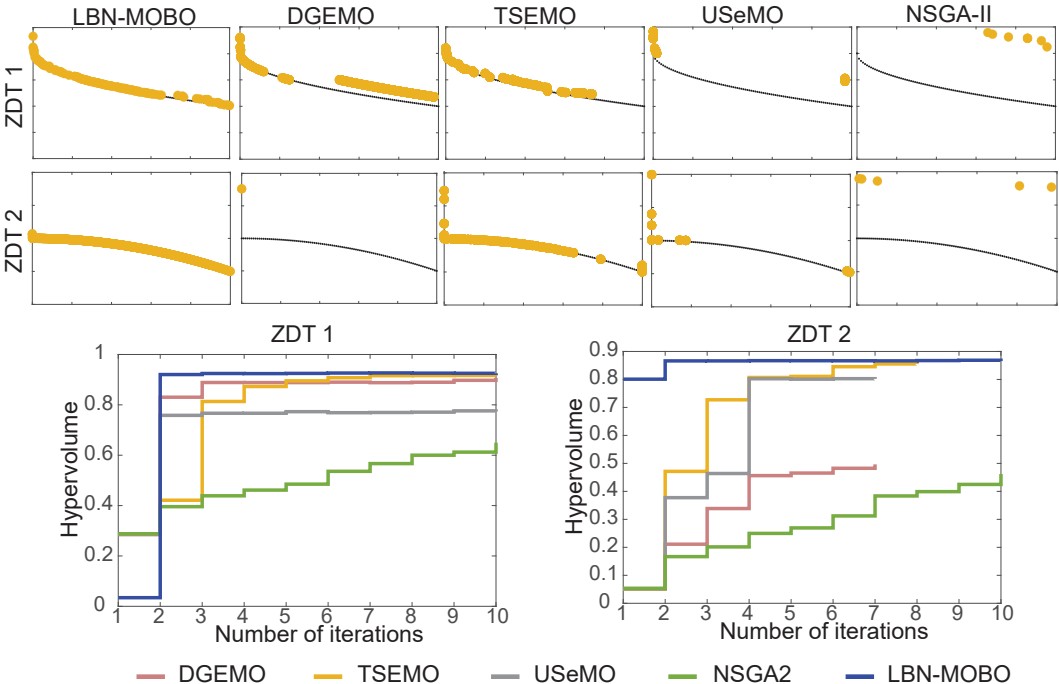

Figure 9: ZDT 1 and ZDT 2 in 30 dimensional setting. Note the immediate convergence of LBN-MOBO to Pareto front (bottom). The Pareto front is presented after 10 iterations of optimization except for USeMO (7 iterations), DGEMO (7 iterations), and TSEMO (8 iterations) in ZDT2 problem.

The Pareto front in Figure 9 for the ZDT1 problem was obtained after 10 iterations for all the methods considered in the analysis. As for the ZDT2 problem, USeMO and DGEMO were able to run for 7 iterations, while TSEMO managed to run for 8 iterations before becoming intractable.

## C.2 DTLZ TEST SUITE WITH 3 DIMENSIONAL OUTPUT

The DTLZ test suite is a popular benchmark set of test problems used for testing the performance of multi-objective optimization algorithms Deb et al. (2005). The DTLZ test suite is characterized by scalable problems, meaning that the number of objectives and decision variables can be easily adjusted. This feature makes it particularly useful for assessing how well algorithms handle problems of varying complexity.

Below are the definitions of the DTLZ1, DTLZ4, and DTLZ5 problems, which we used to benchmark LBN-MOBO for 3D output.

### C.2.1 DTLZ1

DTLZ1 is defined as:

Objective function $f_m$ is given by:

$$f_m(\mathbf{x}) = \frac{1}{2} \left(1 + g(\mathbf{x})\right) \prod_{i=1}^{m-1} (1 - x_i)$$

$$g(\mathbf{x}) = 100 \left( |x| + \sum_{i=1}^{|x|} (x_i - 0.5)^2 - \cos(20\pi(x_i - 0.5)) \right)$$

$$0 \le x_i \le 1, \quad \text{for } i = 1, 2, \ldots, n$$

where $m$ is the number of objectives. Here, $x$ denotes the decision variables vector.

### C.2.2 DTLZ4

DTLZ4 is similar to DTLZ1, but with an additional parameter $\alpha$ that controls the shape of the Pareto front. The objective functions are defined as:

$$f_m(\mathbf{x}) = (1 + g(\mathbf{x})) \prod_{i=1}^{m-1} \cos\left(x_i^\alpha \frac{\pi}{2}\right)$$

$$0 \le x_i \le 1, \quad \text{for } i = 1, 2, \ldots, n$$

where $g(\mathbf{x})$ is the same as in DTLZ1, $m$ is the number of objectives, and $\alpha$ is a user-defined parameter, typically set to 100, which controls the density of solutions.

### C.2.3 DTLZ5

DTLZ5 is formulated to test the algorithm's ability to converge to a curved Pareto front and to maintain diversity among solutions. The DTLZ5 problem is defined as follows:

Objective functions:

$$f_i(\mathbf{x}) = (1 + g(\mathbf{x}_M)) \cos\left(x_1 \frac{\pi}{2}\right) \ldots \cos\left(x_{i-1} \frac{\pi}{2}\right) \sin\left(x_i \frac{\pi}{2}\right), \quad \text{for } i = 1, 2, \ldots, m \quad (7)$$

where

- $\mathbf{x} = (x_1, x_2, \ldots, x_n)$ are the decision variables.
- $m$ is the number of objectives.

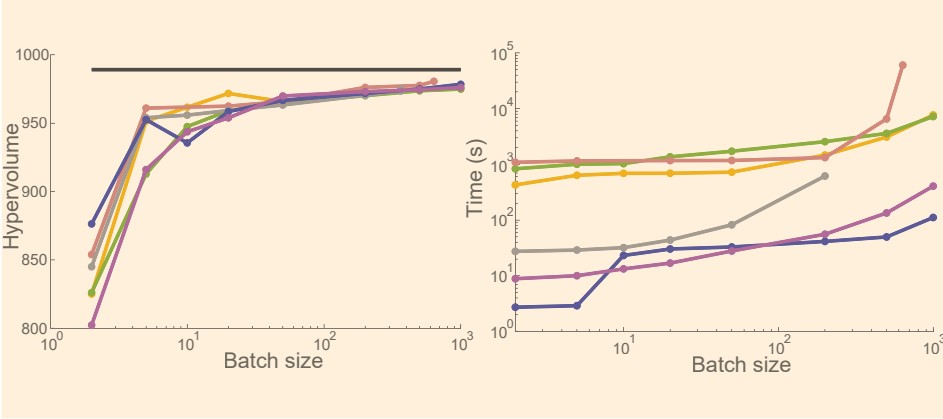

Figure 10: DTLZ5 benchmark optimization via 2$MD$ acquisition function and various neural BO surrogate models. The fitting time of each model was recorded to assess computational efficiency.

- $g(\mathbf{x}_M)$ is a function defined as: $g(\mathbf{x}_M) = \sum_{x_i \in \mathbf{x}_M}(x_i - 0.5)^2$, with $\mathbf{x}_M$ being the subset of decision variables starting from the $m$th variable to the $n$th variable.

The constraints are:

$$0 \le x_i \le 1, \quad \text{for } i = 1, 2, \ldots, n \tag{8}$$

DTLZ5's primary challenge lies in its reduced dimensionality of the search space due to the use of trigonometric functions, which tends to align the solutions along a curve in the objective space. This problem is particularly useful for assessing the ability of an optimization algorithm to handle non-linear relationships between objectives and to generate a well-distributed set of solutions along a curved Pareto front.

### C.3    BENCHMARKING LBN-MOBO AGAINST OTHER BNNs ON DTLZ5 PROBLEM

The DTLZ5 problem, with its extensive input and output dimensions, presents a formidable challenge in optimization. To showcase the robustness and scalability of our pipeline, we applied it to the DLTZ5 test, characterized by 20 input design parameters and 3 output dimensions. As illustrated in Figure 10, this problem displays behavior akin to what we observed in our earlier tests (see Figure 3). Although all the methods we tested show comparable performance, dropout and Deep Ensembles particularly distinguish themselves with their notably higher scalability. The setup of this experiment is identical to the experiment in Section 5.1.

### C.4    BENCHMARKING LBN-MOBO ON DTLZ1 AND DTLZ4 USING A VARIETY OF OTHER OPTIMIZATION TECHNIQUES

In this section, we aim to compare the performance of our method on two problems from the DLTZ test suite Deb et al. (2005) with a 6-dimensional design space and a 3-dimensional performance space. We address both the DLTZ1 and DLTZ4 problems using the same configuration described in the ZDT3 problem Section C.1. during these experiment we have used LBN-MOBO with Deep Ensembles as the surrogate model. Note that for a 3 dimentional output we need to solve a 6 dimentional problem while running our 2MD acquisition function. As the dimensional of the output increases We are going to need more sample budget for our 2MD acquisition function. As a result we solve the DTLZ4 problem using LBN-MOBO with both 1000 and 4000 sample budget.

Figures 11 and 12b illustrates that not only LBN-MOBO has achieved the best Pareto front but also in term of computation time it is by far more scalable than the counterpart methods. Notably, even by increasing the batch size to 4000 samples, which resulted in even better Pareto front, the computation time is still insignificant compared to the rival methods.

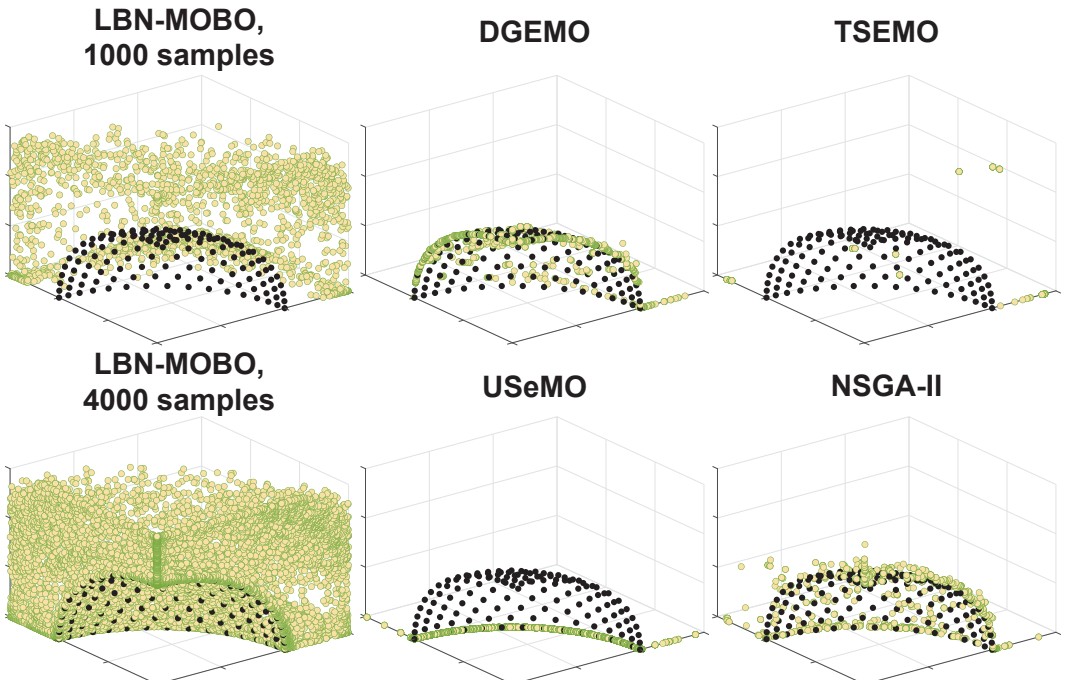

Figure 11: The Pareto front of the DLTZ4 problem with 6-D input and 3-D output.

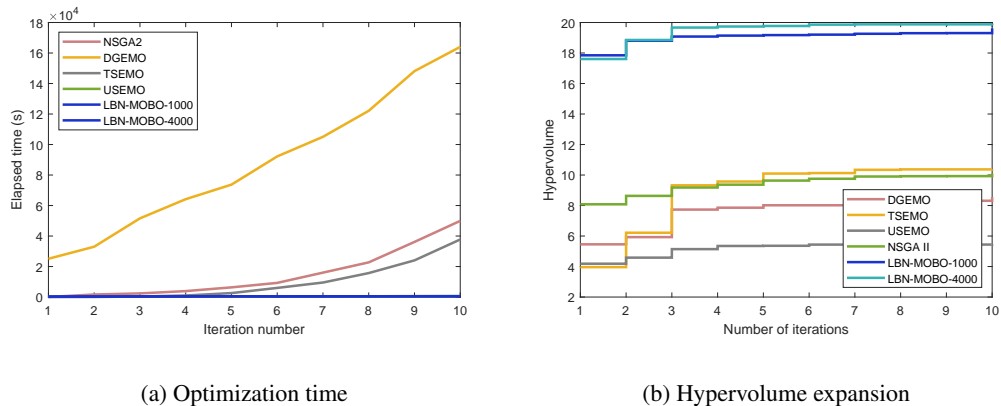

(a) Optimization time

(b) Hypervolume expansion

Figure 12: DTLZ4 experiment with 6 inputs and 3 outputs.

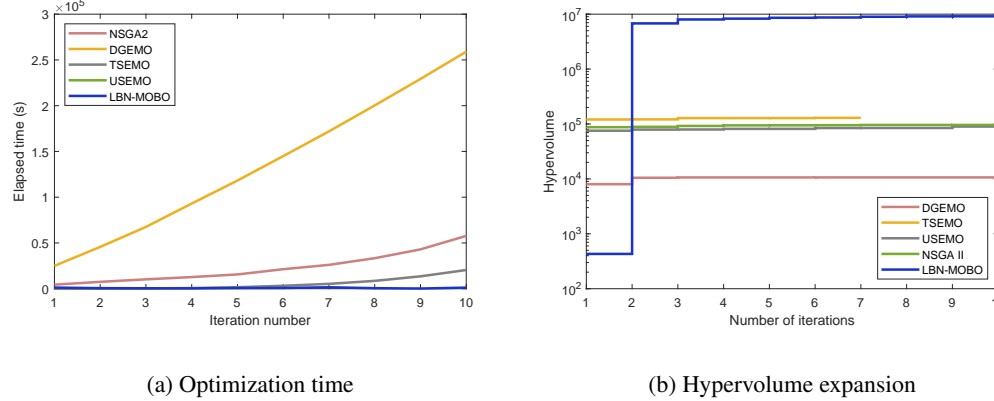

(a) Optimization time           (b) Hypervolume expansion

Figure 13: DTLZ1 experiment with 6 inputs and 3 outputs.

Figure 13b suggests that DLTZ1 problem follows the same pattern except that in this case the achieved Pareto front is orders of magnitude better than the counterpart methods. Likewise from Figure 13a we understand that the computation time for LBN-MOBO is also significantly lower than the other methods.

### C.5 REAL-WORLD EXPERIMENT SET UP

#### C.5.1 AIRFOIL

Airfoil represents, for example, the cross-sectional shape of an airplane wing, with its performance quantified by the lift coefficient $C_L$ and the lift-to-drag ratio $C_L/C_D$ Park and Lee (2010).

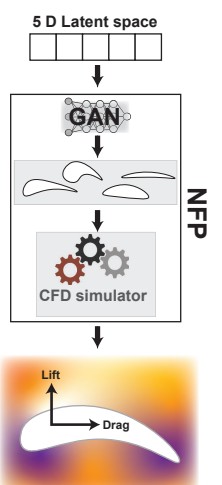

The aim of this experiment is to explore different airfoil shapes to discover the Pareto front of these performances ($C_L$ and $C_L/C_D$). Lift is the upward force that acts perpendicular to the direction of incoming airflow, primarily serving to counterbalance the weight of an aircraft or providing an upward thrust for an airfoil. Drag is the resistance encountered by an object as it moves through a fluid. It acts in the opposite direction to the free stream flow and parallel to it.

Minimizing the drag is important for maximizing the efficiency and speed of vehicles, as well as reducing fuel consumption. In standard computational fluid dynamics (CFD) simulations, the Navier-Stokes equations are solved around the airfoil to compute $C_L$ and $C_L/C_D$. We utilize the open-source software Open-FOAM for running our simulations, setting the free stream angle to 0 and length to 40 OpenFOAM Foundation (2021); Thuerey et al. (2020). The design parameters of this problem describe the shape of the airfoils. Due to high dimensionality and complex shape constraints, we employ a specific type of Generative Adversarial Networks (GANs) to transform the complex design space into a manageable five-dimensional latent space Chen and Ahmed (2021). We assess the shapes generated by GAN using a Computational Fluid Dynamics (CFD) simulator to measure the values of $C_L$ and $C_L/C_D$. As such, our NFP in this problem is a combination of the GAN and the CFD simulator (inset).

#### C.5.2 3D PRINTER'S COLOR GAMUT

A color gamut represents the range of colors that can be achieved using a specific device, such as a display or a printer Wyszecki and Stiles (2000). In this experiment, we compute the color gamut of a 3D printer by determining the Pareto front of a multi-objective optimization problem. A printer combines different amounts of its limited number of inks to create a range of colors. The design parameters of this problem are the amount of available inks. We explore CIEa*b* color space CIE (2004) which is our performance space. CIE a* represents the color-opponent dimension of red-green,

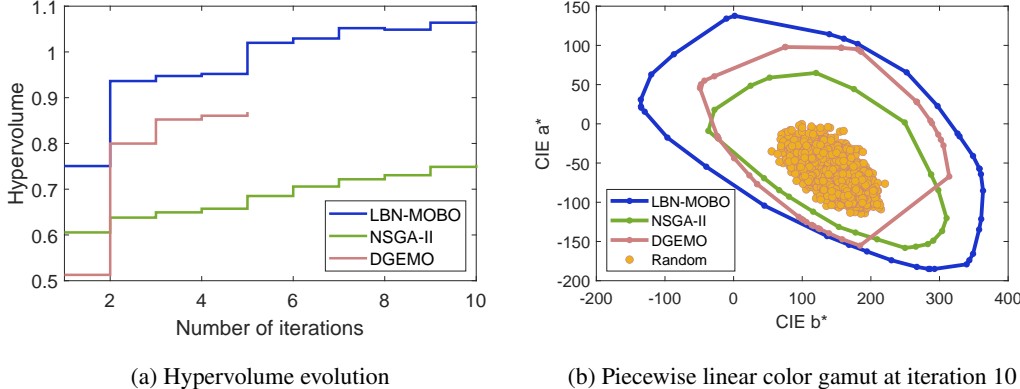

(a) Hypervolume evolution        (b) Piecewise linear color gamut at iteration 10

Figure 14: The hypervolume evolution and gamut of the 44-ink printer calculated by different methods.

with negative values representing green and positive values representing red. CIE b* represents the color-opponent dimension of blue-yellow, with negative values representing blue and positive values representing yellow.

Following Ansari et al. (2022) we create a printer NFP using an ensemble of 10 neural networks (not related to our ensemble surrogate). We create a complex instance of this problem where we simulate a printer NFP with 44 inks Ansari et al. (2021). All networks in the ensemble NFP are trained on 344,000 printed patches with varying ink-amount combinations and their corresponding a*b* colors. This problem has a 44 dimensional design space as the printer NFP assumes 44 inks.

### C.6 COMPLEMENTARY EXPERIMENTS ON THE REAL WORLD PROBLEMS

Following the experiments in Section 5 we compare LBN-MOBO with NSGA-II and DGEMO on our real-world problems. Other than LBN-MOBO, NSGA-II is the sole method capable of managing a batch sizes of 20,000. Additionally, we consider DGEMO due to its competitive performance in ZDT problems (Section C.1), although we must limit its batch size to 1,000 and restrict it to 5 to 6 iterations due to prohibitive run time.

#### C.6.1 EVALUATION OF 44-INK PRINTER GAMUT EXPERIMENT USING NSGA II AND DGEMO

For the task of exploring the 44-ink color gamut, we initialize LBN-MOBO with 10,000 samples, and each subsequent iteration processes a batch size of 20,000 samples. Given the high dimensionality of the design space, this problem poses a significant challenge to many optimization algorithms, making it a fascinating experimental case. The performance space in this experiment is the 2 dimensional a*b* color space. Figure 14a graphically depicts the accelerated increase in hypervolume of the color gamut when using LBN-MOBO. Also, final gamut estimation for NSGA-II and LBN-MOBO after 10 iterations, and DGEMO after 5 iterations, is depicted in Figure 14b, showing a significanlty larger estimated gamut by LBN-MOBO.

#### C.6.2 EVALUATION OF AIRFOIL EXPERIMENT USING NSGA II AND DGEMO

Figure 15 showcases a comparison between NSGA-II, DGEMO, and LBN-MOBO for the airfoil problem. This experiment has a significantly more complex NFP since the relationship between the design and performance space is highly complex as we map the latent code of a GAN to aerodynamic properties. We start LBN-MOBO and NSGA-II with 15,000 samples, and each iteration runs with a batch size of 15,000 all simulated by OpenFOAM, a high-fidelity CFD solver OpenFOAM Foundation (2021).

Once again, as depicted in Figure 15a, LBN-MOBO discovers a superior Pareto front in a remarkably small number of iterations. Although with many more iterations, NSGA-II also reaches an acceptable

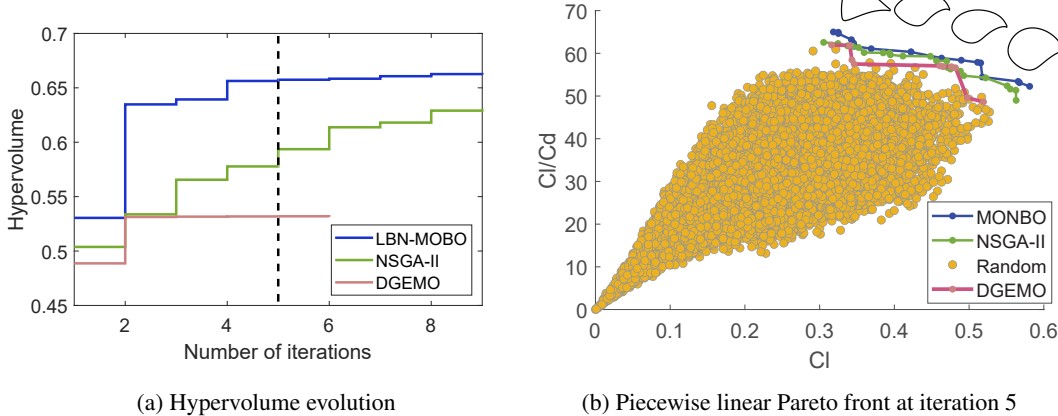

(a) Hypervolume evolution

(b) Piecewise linear Pareto front at iteration 5

Figure 15: The hypervolume evolution and the Pareto front of the airfoil problem of LBN-MOBO and NSGA-II with equal batch sizes.

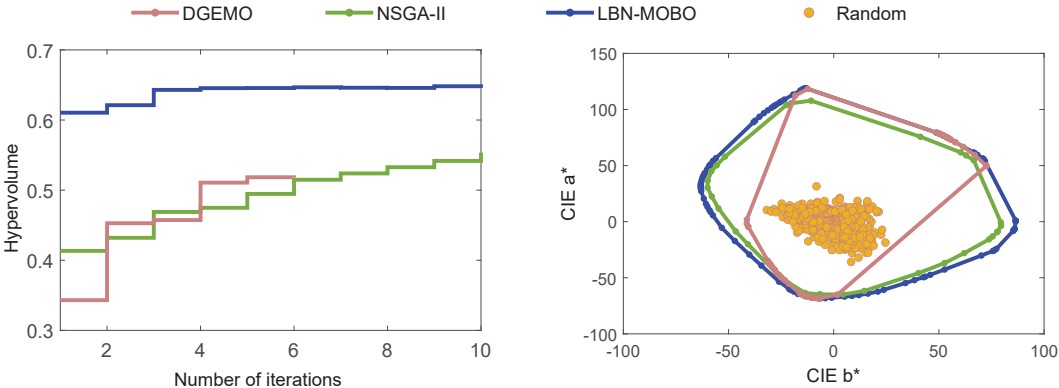

Figure 16: 8-ink printer gamut

Pareto front, this is most likely due to the use of a large batch size for each iteration in a comparatively smaller design space (five dimensions). In contrast, DGEMO's performance significantly deteriorates. This likely stems from its reliance on the precise estimation of the gradient and Hessian of the NFP through the surrogate model. This task becomes increasingly difficult as the complexity of the NFP increases. In Figure 15b we show also the random samples to depict the landscape of peformances.

### C.7 PRINTER COLOR GAMUT FOR 8 INK

Figure 16 displays the results of the 8-ink printer color gamut problem, which has a setup that is nearly identical to the 44-ink problem, with smaller number of available inks and as a result smaller design space. The advantage of this NFP, is in its ability in accurately mimicking the behavior of the Epson printer from which the data set is derived.

As observed, both NSGA-II and DGEMO exhibit results that are closer to LBN-MOBO, which could be attributed to the lower dimensionality of the problem.

## D IMPLEMENTATION DETAILS

### D.1 DEEP ENSEMBLES

For our surrogate model, building on the methodology proposed in Ansari et al. (2022), we have constructed Deep Ensembles, utilizing a diverse collection of activation functions, to enhance the

precision of epistemic uncertainty quantification. This enhancement serves as a cornerstone for ensuring the robust operation of the remaining procedures.

In our implementation, the Deep Ensembles comprises ten sub-networks, each employing a specific activation function as outlined below:

- Tanh $\times 2$ LeCun et al. (2002)
- ReLU $\times 2$ Nair and Hinton (2010)
- CELU $\times 2$ Barron (2017)
- LeakyReLU $\times 2$ Maas et al. (2013)
- ELU Clevert et al. (2015)
- Hardswish Howard et al. (2019)

For the ZDT3 and printer's color gamut networks, we employ a three-layer architecture, with neuron configurations of 100, 50, and 100 per layer respectively. To conduct our analysis on the rival methods, we utilized the pymoo library Blank and Deb (2020) and DGEMO source code Konakovic Lukovic et al. (2020).

Given the complexity inherent to the airfoil problem, it necessitates a more intricately designed network. We configured this network with four hidden layers, containing 150, 200, 200, and 150 neurons, respectively. In the Native Forwarded Process (NFP) of the Airfoil design, i.e., the open source fluid simulator OpenFoam, we have observed that sampling the latent space of the GAN near 0 and, in general, below 0.1 occasionally leads to invalid designs. This occurrence can introduce instability for the optimizers. To address this issue, we have imposed a limitation on the GAN latent space, restricting it between 0.1 and 1 to ensure the generation of valid designs.

Over the course of the LBN-MOBO iterations, data accumulation intensifies. Although it is feasible to progressively enlarge the batch size to maintain a constant total training time, we have chosen to keep the batch sizes fixed due to the minimal increment in training time relative to competing methods. Specifically, we employed a batch size of 20 for the airfoil problem, 10 for ZDT3, and 100 for the printer's color gamut.

All networks underwent a training period spanning 60 epochs.

## D.2 MC DROPOUT

As a general rule of thumb, we opt for wider architectures compared to those in the Deep Ensembles network. This approach is based on the consideration that, since a dropout layer is utilized in every training instance, some of the perceptrons are deactivated, leading to a somewhat narrower sub-network in the trained model. The dropout ratio for all the models is set at 0.05, and ReLU is employed as the activation function. To compute the epistemic uncertainty for every inference, each network is queried 100 times.

For the ZDT3 problem, a network consisting of four fully connected layers with 100, 50, 50, and 100 perceptrons respectively, is trained for 100 epochs. The batch size in this instance is set at 5.

Addressing the printer's color gamut problem, we use a network configuration with three layers, containing 200, 100, and 200 perceptrons. This model is trained with a batch size of 100 for 80 epochs.

Given the Airfoil problem's increased complexity, a network comprising four layers with 300, 400, 400, and 300 perceptrons is employed for modeling. The batch size designated for this problem is 20, and the network is trained for 160 epochs.

## D.3 HARDWARE CONFIGURATION

We leveraged a parallel compute cluster consisting of GPUs and CPUs for the simultaneous training of the network and computation of the acquisition function.

The GPU units within our cluster comprise two models: the NVIDIA Tesla A100, the NVIDIA Tesla A40, and NVIDIA Tesla A16. The GPU units have a memory up to 64GB.

The CPU units in the cluster are AMD EPYC 7702 64-Core Processor.

# E   COMPLEMENTARY DISCUSSIONS

USeMO Belakaria et al. (2020) shares some similarity with LBN-MOBO, however, besides its limitation in handling large batches, its utilization of uncertainty information is much more limited than ours. USeMO finds the Pareto front on their surrogate function by running the NSGA-II Deb et al. (2002). Note that they do not optimize for uncertainty in a simultaneous manner as it happens in our $2M$D acquisition. Instead they use uncertainty in a sequential manner to choose the most promising candidates among the ones already calculated by the NSGA-II. This approach cannot account for the samples that are not Pareto dominant according to performance predictions but have high uncertainty. The results in Figures 7 , 9, 12, 13 in the Appendix also confirm that USeMO is not very successful in recovering a diverse Pareto front.

Similar to USeMO, TSEMO Bradford et al. (2018) utilizes NSGA-II for the calculation of the approximated Pareto set and Pareto front on the computationally inexpensive surrogates. What brings TSEMO closer to our approach is their utilization of Thompson sampling Thompson (1933) to exploit or explore the black box function, guided by the uncertainty information obtained from Gaussian process surrogates.

