# OpenReview forum: "Large-Batch, Iteration-Efficient Neural Bayesian Design Optimization"
_ICLR.cc/2024/Conference — Submitted to ICLR 2024_

### Official Review · Reviewer_2FsD · 2023-10-30

**Soundness:** 4 excellent
**Presentation:** 4 excellent
**Contribution:** 3 good
**Rating:** 8
**Confidence:** 3

**Summary:**

The authors propose a Multi objective optimisation algorithm with a focus on large batch sizes (up to 1000s of points) and few iterations (as low as 10).

Given an objective functions $f:\mathcal{X}\to \mathbb{R}^M$ where $\mathcal{X}\subset \mathbb{R}^d$,  the method is a Bayesian model based algorithm, they propose to use Bayesian neural networks as the surrogate model to predict $\underline{y}=\hat{f}(x)$ as well as the epistemic uncertainty $Var(\hat{f(x)})$, these models which can scale to large dataset sizes much more effectively than the more traditional Gaussian processes.

In order to determine a new batch of points to be evaluated, the authors propose to concatenate the model predictions and uncertainties $[\hat{f(x)}, Var(\hat{f}(x))] \in \mathbb{R}^{2M}$, which can then be fed into NSGA-II, a popular evolutionary aglgorithm, which can find the set of pareto optimal points $x_1,...,x_B\in \mathcal{X}$ that form the pareto front in the augmented output space $\mathbb{R}^{2M}$. In other words, these are point that are predicted to have high value and/or high uncertainty.

The authors perform experiments with a range of off-the-shelf Bayesian neural network methods and determine Deep Ensembles to be the best candidate surrogate model.

**Strengths:**

- Simplicty, elegance.
  - Bayesian neural networks have become a work horse surrogate model in the Bayesian Optimization community in recent years
  - NSGA-II is a very popular well established mainstream algorithm in the multi objective community
  - concatenating predictions and uncertainties to be fed into NSGA-II seems a very reasonable good idea
  - altogether the method avoids introducing any sophisticated new engineering, and instead opts to intelligently combine established components from the community with some well justified tweaks.

- clearly written, I enjoyed the exposition of related work.

- Section 5.4, running algorithm without using uncertainties I felt was a very nice experiment and cleawrly demonstrated their benefit.

**Weaknesses:**

I only have minor comments

- I see the authors discuss this in Appendix E but MOO is an very large field and I would be very surprised if batch construction by finding the pareto front of concatenated predictions and uncertainties has not been considered before, (it _seems_ so obvious!),
- upon first reading, I felt the title was somewhat cluttered.

**Questions:**

- presumably for small use cases, optimising 2 simple objectives over 2 dimensions batchsize 2, i.e. the ideal use case for any GP-BO, the proposed method would suffer, is there a crossover from where more simple GP-BO methods fail and LBN-MOBO would be best?

---

> ### Author Response · Authors · 2023-11-21
>
> Thank you for your feedback. We are heartened to know that you appreciate our method’s simplicity and its potential impact. We provide responses to the raised questions below.
>
> ## Is there a crossover from where more simple GP-BO methods fail and LBN-MOBO would be best?
> In our empirical studies on batch multi-objective Bayesian Optimization (BO), we've identified that a batch size of 100 serves as a threshold, beyond which traditional BO methods begin to encounter difficulties. Interestingly, there are some ingenious adaptations to BO that enable operation with batch sizes approaching 1000. However, for even larger batch sizes, the shift towards Bayesian Neural Networks (BNNs) in place of Gaussian Processes (GPs) and replacing the acquisition function with the 2MD function becomes essential. As we scale up to batch sizes of 10,000, only methods like Monte Carlo Dropout (MCDropout) and Deep Ensembles in combination with 2MD acquisition prove capable of effectively handling such extensive batch sizes.
> ## New title.
> We are open to consider renaming the paper and we appreciate to know your feedback in this regards. We think we can drop ‘iteration-efficient’ and ‘design’ from the title.

---

### Official Review · Reviewer_9JiQ · 2023-10-30

**Soundness:** 2 fair
**Presentation:** 2 fair
**Contribution:** 2 fair
**Rating:** 5
**Confidence:** 3

**Summary:**

The paper considers a setting in which BO is applied to solve black-box optimization problems where there are multiple objectives and a query is expensive, but the batch size can be extremely large. To address this challenge, the authors propose an acquisition function that is more scalable and takes into account the uncertainty of candidates. Empirically, this BO algorithm is applied to solve two realistic black-box optimization problems in this setting.

**Strengths:**

1.	The paper considers a novel black-box optimization setting where there are multiple objectives and the batch size can be very large. The authors empirically observe that contemporary multi-objective batch acquisition functions do not scale well with respect to the batch size.
2.	To solve this issue, the authors propose a modified version of Deep Ensembles to approximate BNN and an acquisition function to maximize both predicted objectives and the uncertainty measure.

**Weaknesses:**

1.	Contribution is not enough. The innovation of the paper can be summarized into a new predictive model with a minor modification on the original Deep Ensembles model and a new multi-objective batch acquisition function. For the predictive model, the reason for modifying the uncertainty measurement part from aleatory noise to epistemic noise is unclear. Also, the benefit from this change is not verified in the paper. Moreover, the novelty of 2$\textit{M}$D acquisition compared to the other acquisition functions is not clear either, except for being more scalable.
2.	The empirical results presented are somewhat unconvincing. The reason why the authors choose deep ensembles as a surrogate to test in the two subsequent realistic tasks is that its runtime is shorter and achieves higher hypervolume. However, since this appears in only one experiment, its generalized performances to other tasks are not necessarily better than other surrogates.
3.	More benchmark models should be considered in the two realistic tasks. In these two tasks, only two models are considered, i.e., modified deep ensemble + 2$\textit{M}$D and dropout + 2$\textit{M}$D. Therefore, whether the modified deep ensemble + 2MD indeed performs well enough is not clear. It would be great if the authors can also consider more models as benchmarks.
4.	Limited theory is developed for this new BO method.

**Questions:**

1.	How is the “time” defined in Figures 1 and 3?
2.	How well does the modified deep ensemble quantify uncertainty, compared to the original deep ensemble?
3.	How does 2MD work? It looks like a key ingredient of the new acquisition is the NSGA-II. However, this is not explained in the main body of the paper. Also, how to implement the acquisition function, one of the most important parts in this paper, is not clearly explained. The only relevant statement is the last three lines in page 6.
4.	In reality, the magnitude or the range of $F$ is usually unknown. How do the authors suggest to balance the tradeoff between the output of the predictive model and the uncertainty?

**Details Of Ethics Concerns:**

NA.

---

> ### Author Response · Authors · 2023-11-21
>
> Thank you for the thorough and thoughtful review. We appreciate the detailed feedback. We provide responses to the raised questions below.
>
> ## The reason for modifying the uncertainty measurement part to include only epistemic uncertainty and its benefits.
> Uncertainty in BO primarily serves as a guide for effective exploration. Epistemic uncertainty represents a lack of knowledge about a phenomenon, which can be mitigated through additional observations. Conversely, aleatoric uncertainty relates to the inherent noise in the system and is not a function of data accumulation.
>
> In our study, we worked with the underlying Native Forward Processes (NFPs) that are essentially noiseless. Consequently, addressing noisy NFPs is earmarked for future exploration (Limitation section). Given this noiseless scenario, aleatoric uncertainty is presumed negligible, leading us to focus solely on the epistemic uncertainty. This decision significantly simplified and stabilized our model's training phase. This means our training process closely resembles that of conventional neural network, avoiding potential complexities and instabilities in training Deep Ensembles [1][2]. This methodological choice not only enhances the robustness of LBN-MOBO but also increases its accessibility and practicality as any neural net architecture can be utilized.
>
> ## Usefulness of aleatoric uncertainty.
> Prompted by the reviewer's insightful question, we explored a toy noisy NFP to see whether considering aleatoric uncertainty helps with the exploration. This example shows that in the presence of significant noise, using only the epistemic uncertainty could encourage sampling of noisy regions (likely due to Deep Ensembles mechanism), which will deteriorate the exploration. Very interestingly, including the aleatoric uncertainty and avoiding the regions with such uncertainty improves the exploration significantly. This has highly appealing implications for BO in the presence of noise and we plan to study it intensively in the future.  This is detailed in our supplementary material (Rebuttal PDF_LBN_MOBO_noisy.pdf).
>
> These findings highlight the potential challenges and benefits of aleatoric uncertainty in scenarios with inherent noise, paving the way for future research in this domain.
>
> ## Comparing 2MD acquisition to the other acquisition functions, except for being more scalable.
> The traditional approaches to acquisition, such as the Upper Confidence Bound (UCB), relies on some form of weighted sum of uncertainty and prediction. This weighting is crucial as it dictates the balance between exploration and exploitation within the optimization process. However, the scale differences between prediction and uncertainty necessitate careful tuning of this hyperparameter. Additionally, as the optimization progresses, the requirements for exploration or exploitation may change, potentially requiring ongoing adjustments to this weight. In the realm of multi-objective optimization, the complexity escalates further due to the need to fine-tune multiple parameters to achieve an optimal balance in exploring various objectives and their respective uncertainties.
>
> In contrast, the 2MD acquisition function, as described in the paper (Section 4.2 and Equation 3), adopts a fundamentally different approach. Here, uncertainties and predictions are treated as separate optimization objectives, which are then jointly optimized to construct a Pareto front of potential candidates. This method does not need the summation of objectives, thereby remedies the issue of scale mismatch between objectives. Consequently, our approach is free from the constraints of hyperparameter tuning, simplifying the optimization process. It's important to note that, to our knowledge, there are no alternative methods capable of scaling to the large batch sizes required for addressing real-world problems of the magnitude we are tackling, positioning our algorithm as a unique solution in this domain.
>
> ## More benchmark algorithms other than BNNs on real problems.
> In appendix section C5 you can find complementary experiments comparing LBN-MOBO using other algorithms that are partially or completely capable of handling such large batch sizes.
>
> ## How is the “time” defined in Figures 1 and 3?
> It is the summation of acquisition time plus training the surrogate (a BO cycle) for 10 iterations.
>
> ## How well does the modified deep ensembles quantify uncertainty, compared to the original deep ensemble?
> There is a widely established way of implementing Deep Ensembles where the training of prediction networks is separated from the uncertainty estimation networks [2]. The proposed simplified implementation is identical to that approach when it comes to computing the epistemic uncertainty: we ensemble prediction networks to calculate the epistemic uncertainty.

---

> > ### Author Response · Authors · 2023-11-21
> >
> > ## More experiments to show the advantage of our proposed surrogate over its possible alternates
> > We have included another example with larger input and output dimensions to demonstrate the robust performance and scalability of our pipeline. We used the DLTZ5 test, which has 20 input design parameters and 3 output dimensions, providing a rigorous challenge. The Section C3 in the revised manuscript shows that this problem behaves similarly to our previous tests. While the performance of all methods is comparable, dropout and Deep Ensembles stand out for their significantly higher scalability.
> >
> > To align these experiments more closely with our real-world scenarios, we conducted additional tests using HMC and SGHMC methods (the two BNNs that could scale beyond 1000 sample batch sizes) with large batch sizes of 5000 and 10000 on the 30-dimensional ZDT3 problem (in a setup similar to Sections 5.1 and C1.4). The results, including runtime and objective values, were captured within a 24-hour GPU time frame.
> >
> > | Method         | Batch Size | Objective | Runtime  |
> > |----------------|------------|-----------|----------|
> > | HMC            | 5000       | 0.2217    | 53433s   |
> > | HMC            | 10000      | Failed    | Failed   |
> > | SGHMC          | 5000       | 0.1921    | 43749s   |
> > | SGHMC          | 10000      | Failed    | Failed   |
> > | DeepEnsembles  | 5000       | **0.3001**    | **721s**     |
> > | DeepEnsembles  | 10000      | **0.3001**   | **1377s**    |
> >
> > The data in our table illustrates that the Deep Ensemble not only achieves a better Pareto front but also does it much faster compared to other BNNs. Furthermore, the table highlights the impracticality of using other BNNs in the context of our real-world problems, as we employed batch sizes exceeding 10,000 samples. This finding reinforces the effectiveness and efficiency of our chosen methods in handling large-scale, complex problems.
> >
> > ## More details on the implementation of 2MD acquisition function and the role of NSGA-II.
> > 2MD acquisition treats the surrogate model as a black box function and perform the acquisition by iteratively querying the surrogate while using the NSGA-II algorithm to guide the system toward the Pareto front of the the best solutions and highest epistemic uncertainties. These solutions are the candidates that we evaluate using the NFP to generate the next-generation dataset.
> >
> > NSGA-II serves as a scalable multi-objective optimization inside the pipeline. If for a specific task one can come up with a more suitable optimizer, NSGA-II can easily be replaced. However since querying the Deep Ensembles surrogate is very inexpensive, NSGA-II proves to be a very effective and cost-efficient choice in practice.
> > To gain more information about its mechanism inside LBN-MOBO pipeline, please refer to our code: https://github.com/an-on-ym-ous/lbn_mobo
> >
> > ## How to balance the trade-off between the output of the predictive model and the uncertainty?
> > In the 2MD acquisition function, uncertainties and predictions are handled as independent objectives. We optimize uncertainties and predictions as different objectives simultaneously to create a Pareto front of possible choices. Our method avoids combining the two, which means we don't have to worry about their different magnitudes. As a result, our method doesn't need complicated hyperparameter tuning, making the optimization process simpler.
> >
> > [1] Seitzer, Maximilian, et al. "On the pitfalls of heteroscedastic uncertainty estimation with probabilistic neural networks." arXiv preprint arXiv:2203.09168 (2022).
> >
> > [2] Nix, David A., and Andreas S. Weigend. "Estimating the mean and variance of the target probability distribution." Proceedings of 1994 ieee international conference on neural networks (ICNN'94). Vol. 1. IEEE, 1994.

---

> > > ### Comment · Reviewer_9JiQ · 2023-11-23
> > > **Response to the Authors**
> > >
> > > I appreciate the authors for their efforts on the additional experiments and detailed explanations. My concerns are mostly addressed. I raised my score to 5.
> > >
> > > Nevertheless, I still think the paper could benefit from a deeper theoretical interpretation of why the proposed algorithm should work better than others, especially, on the part of only modeling epistemic uncertainty versus the traditional deep ensembles. Moreover, the absence of tests on the algorithms’ performance in noisy environments, a more common real-world scenario, limits its credibility on practical applicability and impact.

---

> ### Author Response · Authors · 2023-11-23
>
> We appreciate the reviewer's openness and for raising our score. The answer below will hopefully addresses the remaining concerns of the reviewer.
> ## The benefits of using only epistemic uncertainty in noiseless NFPs and the advantages of aleatoric uncertainty in noisy settings.
> Given your feedback and the one of Reviewer acfL, we conducted additional experiments to study the effect of including the aleatoric uncertainty in our acquisition function in noiseless NFPs. These experiments are shown in the appended document (Rebuttal PDF_LBN_MOBO_aleatoric.pdf). We used the examples and procedures described in Sections C1.4 and C1.5 of the paper. The outcomes indicate that incorporating aleatoric uncertainty does not affect the exploration process in a beneficial manner—as showcased by the ZDT1 results—and may even delay the convergence, as observed in the ZDT2 and ZDT3 experiments. This is attributed to the nature of the aleatoric uncertainty, that measures inherent noise which cannot be diminished. This follows our original intuition that aleatoric uncertainty cannot be an effective tool for driving the exploration.
>
> **However**, the noisy example detailed in our supplementary document (PDF-LBN_MOBO_noisy.pdf) demonstrates that for noisy Native Forward Processes (NFPs), incorporating aleatoric uncertainty is advantageous for _avoiding_ (and not exploring) the system's irreducible noise during BO. Motivated by these findings, we are applying LBN-MOBO with both epistemic and aleatoric uncertainties on our printer’s color gamut problem where we intentionally add noise to one the printer channels (in simulation). This new experiment will be incorporated into the revised version of our paper.

---

### Official Review · Reviewer_s2Pz · 2023-10-31

**Soundness:** 3 good
**Presentation:** 2 fair
**Contribution:** 2 fair
**Rating:** 6
**Confidence:** 4

**Summary:**

The paper considers the problem of designing Bayesian optimization algorithms for the setting of large batches of evaluations in order to optimize a black-box function. An acquisition function is constructed as multiobjective optimization over multiple predictive mean and uncertainty functions modeled by a deep ensemble. Experiments are performed on two real-world benchmarks.

**Strengths:**

- The paper considers an important problem relevant to real-world applications in engineering design.

- I especially like the real world evaluation on two interesting benchmarks: airfoil design and 3D printing. It would be an interesting contribution to the BO community if they are released in the open-source code.

- The idea is simple and works well on the benchmarks.

**Weaknesses:**

- Although I like the simplicity of the approach, the reasoning behind choosing this instantiation of multiobjective optimization is not entirely clear. Please considering some more analysis about the principles behind the proposed acquisition function.

- Some relevant related work that can be useful to discuss in the paper:
	- A very similar idea utilizing multiobjective acquisition function with predicted mean and variance as objectives.

	[1] Gupta, S., Shilton, A., Rana, S., & Venkatesh, S. (2018, March). Exploiting strategy-space diversity for batch Bayesian optimization. In International conference on artificial intelligence and statistics (pp. 538-547). PMLR.
	- There has been a bunch of work on making thompson sampling work for large batch sizes in both continuous and combinatorial design spaces.

	[2] Hernández-Lobato, J. M., Requeima, J., Pyzer-Knapp, E. O., & Aspuru-Guzik, A. (2017, July). Parallel and distributed Thompson sampling for large-scale accelerated exploration of chemical space. In International conference on machine learning (pp. 1470-1479). PMLR.

	[3] Deshwal, A., Belakaria, S., & Doppa, J. R. (2021, May). Mercer features for efficient combinatorial Bayesian optimization. In Proceedings of the AAAI Conference on Artificial Intelligence (Vol. 35, No. 8, pp. 7210-7218).

	[4] Vakili, S., Moss, H., Artemev, A., Dutordoir, V., & Picheny, V. (2021). Scalable Thompson sampling using sparse Gaussian process models. Advances in neural information processing systems, 34, 5631-5643.

- Probably a nit, but I think calling deep ensembles as a bayesian neural network is not entirely correct.

**Questions:**

Please see weaknesses section above.

---

> ### Author Response · Authors · 2023-11-21
>
> Thank you for your insightful feedback. Your appreciation of our work, particularly the real-world evaluations and the potential contribution to the Bayesian Optimization community, is highly encouraging. Following your suggestion to release our benchmarks in open-source code, we will certainly consider this moving forward. We provide responses to the raised questions below.
>
> ## More analysis on 2MD acquisition function.
> We have provided the following insights and analyses about our acquisition function:
> - In our paper, Section 5.4 is dedicated to analyzing the influence of uncertainty on the performance of the 2MD acquisition function. Through this analysis, we aim to emphasize the critical role that epistemic uncertainty plays in determining the overall quality of the LBN-MOBO approach.
>
> - We have conducted a preliminary evaluation on the feasibility of applying the 2MD Pareto front approach to noisy Native Forward Processes (NFPs). This examination, detailed in a separate supplementary material (Rebuttal PDF-LBN_MOBO_noisy.pdf), explores the impact of incorporating aleatoric uncertainty into the acquisition function. Our findings reveal that relying solely on epistemic uncertainty in noisy scenarios can inadvertently lead the optimization process to select candidates from these noisy regions. In contrast, by taking into account aleatoric uncertainty and striving to minimize it during the optimization process, we can effectively avoid these noisy areas. This insight highlights the importance of considering both types of uncertainty in the optimization strategy when dealing with noisy NFPs.
>
> - We have introduced a new subsection that delves into how the 2MD Pareto front can be paired with any arbitrary acquisition function. This aspect is crucial as it positions 2MD as a generic acquisition function adaptable to various contexts. In Section 5.1, we demonstrated this adaptability by coupling the 2MD approach with a range of Bayesian Neural Networks (BNNs).
>
> - In Section C3, we revisited the evaluations from Section 5.1, this time using a new problem, i.e., the DTLZ5 test suite. This problem has 20 input and 3 output dimensions. (When applying 2MD acquisition function, this translates to a 6D problem.) Although Monte Carlo Dropout (MCDropout) and Deep Ensembles showed the best runtime performance, the quality of the solutions across all surrogates was highly competitive. This finding suggests that the 2MD acquisition approach is capable of delivering strong performance with various surrogate models, indicating its versatility and effectiveness in handling complex multi-dimensional optimization tasks.
>
>
> ## Expansion of complementary related work.
> We appreciate the reviewer for mentioning these missing related works. We have included them in the complementary related work in Sections A2 and A3.

---

### Official Review · Reviewer_acfL · 2023-11-01

**Soundness:** 2 fair
**Presentation:** 2 fair
**Contribution:** 2 fair
**Rating:** 3
**Confidence:** 4

**Summary:**

This paper presents a new method to perform multi-objective BO with the large-batch setting. It proposes to use a Bayesian Neural Network (BNN) created by Deep Ensembles as a surrogate model. It also proposes an NSGA-II based acquisition function that is claimed to be able to scale to large-batch setting better than current acquisition functions. The main idea for the proposed acquisition function (2MD acquisition function) is to simultaneously maximize the predicted objectives and the associated uncertainties, both are given by the BNN surrogate model.

The method (LBN-MOBO) is evaluated on synthetic functions (1 in the main paper and 4 in the appendix), and 2 real-world problems.

**Strengths:**

- The paper tackles an important problem which is performing multi-objective BO with the large-batch setting.
- The paper proposes an acquisition function for applying large-batch when performing BO while the current acquisition functions (qEHVI, qParEGO, qNEHVI) struggle, in terms of computation time. The concept of the proposed acquisition function is intuitive: it seems to further encourage explorative behavior, because it also maximizes the uncertainties in the surrogate model.

**Weaknesses:**

- Some technical details are not described clearly, making it sometimes hard to catch the main idea of the paper. For example, the formal problem statement is not described, the proposed method makes use of only epistemic uncertainty but the concept of epistemic uncertainty is not explained in the Background. The organization of the paper is sometimes a bit confused, for example, the overall process of BO should not be placed in method section.
- The use of BNN as the surrogate model to enhance the performance is surely promising, however, BNN has many problems. For example, the tuning of its hyperparameters could be another optimization problem or the uncertainty provided by the BNNs could be inaccurate. However, these problems are not discussed in the paper. Furthermore, I don't understand why the proposed method only requires the epistemic uncertainty. There are no motivation, explanation, or insights about this choice and why it work.
- The proposed acquisition function of optimizing both the prediction and the uncertainty and the usage of NSGA-II to optimize this acquisition function seems to be not too novel for me. The idea is very similar to UCB. There are no deep analysis regarding this proposed acquisition function and why it will work well.
- The experimental evaluation is very limited. It doesn't compare against other baselines in the main paper. In Section 5.1, it is not convincing to choose the surrogate model (inference method) by using only 1 synthetic experimental result.
- Related works should mention other types of surrogate models apart from GP and BNN, such as TPE and RF. And also, it is worth mentioning why BNN is preferred over these models.
- Section 4.1 only covers the modification for Deep Ensembles method. It is not clear how to apply the modification for other inference methods (SGHMC, HMC, DKL, IBNN), so as to compare in Figure 3.

Minor:
- Authors should use \citep{} and \citet{} separately when citing the references.

**Questions:**

Apart from my comments in the Weaknesses section, the authors can answer the following questions:
- The concept of the 2MD acquisition function is quite similar to Upper Confidence Bound with a specific exploration factor. It seems that in UCB, both the prediction and the uncertainty are incorporated to compute the acquisition function, while 2MD use the two values as separate objective to optimize. Can the authors point out some differences between the UCB and 2MD?
- In Figure 1, why there is no surrogate SGHMC, HCM, Deep Ensembles paired with qNEHVI and qParEGO. How many function evaluations in total for this experiment?
- Is batch size b > 1000 a normal batch size in real-world problem? There seems to be no reference to any applications using such large batch.
- The two real-world problems use b=15,000 for airfoil problem and b=20,000 for printer problem, on a total of 10 iterations. With such a large number of function evaluations (150,000 and 200,000), can LBN-MOBO outperform Evolutionary Computation methods, e.g., MOEA/D, NSGA-II? These two EC methods are quite powerful for solving multi-objective optimization problems.

[1] B. Paria, K. Kandasamy, and B. Póczos. A flexible framework for multi-objective Bayesian optimization using random scalarizations. In Proceedings of The 35th Uncertainty in Artificial Intelligence Conference, volume 115, 2020

[2] Daulton, Samuel, David Eriksson, Maximilian Balandat, and Eytan Bakshy. "Multi-objective bayesian optimization over high-dimensional search spaces." In Uncertainty in Artificial Intelligence, pp. 507-517. PMLR, 2022.

---

> ### Author Response · Authors · 2023-11-21
>
> Thank you for the thorough and thoughtful review. We appreciate the raised questions that help us clarify the paper. We provide responses to the raised questions below.
>
> ## Explanation of the epistemic uncertainty is missing.
> We have elaborated on epistemic and aleatoric uncertainty and their differences in Section 4.1.
>
>
> ## Why does LBN-MOBO only needs epistemic uncertainty (and not aleatoric uncertainty)?
> Epistemic uncertainty represents the lack of knowledge about a phenomenon and can be reduced with more observation. The aleatoric uncertainty is a sign of inherent noise which cannot be reduced. In this paper we assumed that our underlying Native Forward Processes (NFPs) do not feature significant noise and left handling noisy NFPs for the future work (See Limitation section). In the absence of significant noise, the inclusion of aleatoric uncertainty does not affect the solutions. Thus, we decided to modify the Deep Ensembles to calculate only the epistemic uncertainty which is necessary for the exploration. This decision significantly simplifies and stabilizes our model's training [1] phase, as we don’t need to train variance networks typically needed for aleatoric uncertainty [2].
>
> Prompted by the reviewer's insightful question, we explored a toy noisy NFP to see whether considering aleatoric uncertainty helps with the exploration. This example shows that in the presence of significant noise, using only the epistemic uncertainty could encourage sampling of noisy regions (likely due to Deep Ensembles mechanism), which will deteriorate the exploration. Very interestingly, including the aleatoric uncertainty and avoiding the regions with such uncertainty improves the exploration significantly. This has highly appealing implications for BO in the presence of noise and we plan to study it intensively in the future.  This is detailed in our supplementary material (Rebuttal PDF_LBN_MOBO_noisy.pdf).
>
> ## Discussion of the problems inherent to BNNs (e.g., tuning and uncertainty quality).
> The reviewer is right about these challenges of BNNs. While we don’t focus on improving on the BNNs side, our simplified approach in Sections 4.1 and 4.2 addresses both concerns to a great extent. Regarding the surrogate model, by excluding the aleatoric uncertainty we have avoided several complexities and instabilities during the training phase [1][2], making it completely similar to training a group of conventional neural networks. Moreover the way we calculate the epistemic uncertainty is identical to the original Deep Ensembles methods and it is among the best (quasi-)BNNs in terms of the trade off between the quality of uncertainty and scalability [3].
>
> ## Similarity and differences between UCB and 2MD Pareto front.
> As reviewer correctly points out UCB and 2MD perform the optimization while exploiting the uncertainties in different ways.
> UCB works based on a weighted sum of uncertainty and prediction. In UCB tuning is important as it regulates the trade-off between exploration and exploitation. Given the magnitude difference between prediction and uncertainty, one needs to tune this hyperparameter. Moreover, as we progress through the optimization the need for exploration or exploitation might change and as a result we might need to constantly adapt this hyperparameter. In the case of multi-objective optimization, the number of tuning parameters increases as we need to tune and find the right balance in exploring different objectives in the presence of their uncertainties.
>
> In the 2MD acquisition function, however, the uncertainties and predictions remain independent objectives and are jointly optimized to generate a Pareto front of viable candidates (Section 4.2 and Equation 3). Since each objective is optimized jointly with other objectives without any form of summation we do not need to worry about the scale difference and LBN-MOBO does not introduce extra hyperparameters.
>
> Unfortunately, we could not find any implementation of UCB extension capable of handling both multi-objective and batch optimization to compare it against our method. If pointed to such an implementation, we will be happy to compare it with our 2MD algorithm. To the best of our knowledge, qEHVI and qNEHVI are the state-of-the-art algorithms for multi-objective batch Bayesian optimization. We showed that these methods are not suitable for the large batch setting we are proposing (Section 3).

---

> ### Author Response · Authors · 2023-11-21
>
> ## More examples to show the advantage of our proposed surrogate over its possible alternatives.
> We have included another example with larger input and output dimensions to demonstrate the robust performance and scalability of our pipeline. We used the DLTZ5 test, which has 20 input design parameters and 3 output dimensions, providing a rigorous challenge. The Section C3 in the revised manuscript shows that this problem behaves similarly to our previous tests. While the performance of all methods is comparable, dropout and Deep Ensembles stand out for their significantly higher scalability.
>
> To align these experiments more closely with our real-world scenarios, we conducted additional tests using HMC and SGHMC surrogates (the two BNNs that could scale beyond 1000 sample batch sizes) with large batch sizes of 5000 and 10000 on the 30-dimensional ZDT3 problem (in a setup similar to Sections 5.1 and C1.4). The results, including runtime and objective values, were captured within a 24-hour GPU time frame.
>
> | Method         | Batch Size | Objective | Runtime  |
> |----------------|------------|-----------|----------|
> | HMC            | 5000       | 0.2217    | 53433s   |
> | HMC            | 10000      | Failed    | Failed   |
> | SGHMC          | 5000       | 0.1921    | 43749s   |
> | SGHMC          | 10000      | Failed    | Failed   |
> | DeepEnsembles  | 5000       | **0.3001**    | **721s**     |
> | DeepEnsembles  | 10000      | **0.3001**   | **1377s**    |
>
> The data in our table illustrates that the Deep Ensemble not only achieves a better Pareto front but also does it so much faster compared to other BNNs. Furthermore, the table highlights the impracticality of using other BNNs in the context of our real-world problems, as we employed batch sizes exceeding 10,000 samples. This finding reinforces the effectiveness and efficiency of our chosen methods in handling large-scale, complex problems.
>
> ## Implementation details of using 2MD Pareto front with other BNNs
> In principle, one of the main advantages of 2MD acquisition function is that it only queries the surrogate models (without the need of gradients for example). 2MD performs by iteratively using these queried information and evolves the solution until a practically good convergence. We included a section and a conceptual figure in the appendix to clarify how it works (Section B2).
>
> ## Is batch size b > 1000 a normal batch size in real-world problem?
> Absolutely! Specifically in the contex of manufacturing, fabrication and simulation (for which we have shown two example experiments in the paper). In many such cases we are capable of producing very large batches of data with almost the same cost of a single sample but iterative lab visits are costly. As a result, LBN-MOBO makes the overall process efficient by reducing the necessity of lab visits (second paragraph of the introduction).
>
> ## Can LBN-MOBO outperform Evolutionary Computation methods?
> Yes. In section C1.4 of appendix we have compared LBN-MOBO against NSGA-II and a few other advanced (not necessarily neural) BO methods on ZDT1, ZDT2, ZDT3, DLTZ1, and DLTZ4 test suits as well as on **both our real-world problems** (Sections C5.1 and C5.2). In all these experiments LBN-MOBO outperforms all other methods handily.
>
> ## Why qEHVI and qNEHVI do not include BNNs other than IBNN and DKL?
> Please refer to the final paragraph of section 3.
>
> [1] Seitzer, Maximilian, et al. "On the pitfalls of heteroscedastic uncertainty estimation with probabilistic neural networks." arXiv preprint arXiv:2203.09168 (2022).
>
> [2] Nix, David A., and Andreas S. Weigend. "Estimating the mean and variance of the target probability distribution." Proceedings of 1994 ieee international conference on neural networks (ICNN'94). Vol. 1. IEEE, 1994.
>
> [3] Lakshminarayanan, Balaji, Alexander Pritzel, and Charles Blundell. "Simple and scalable predictive uncertainty estimation using deep ensembles." Advances in neural information processing systems 30 (2017).

---

> > ### Comment · Reviewer_acfL · 2023-11-22
> >
> > Hi author(s),
> >
> > Thank you for your response. The response has addressed some of my concerns, but my main concerns still remain.
> >
> > The first concern is regarding the choice of using only epistemic uncertainty (and not aleatoric uncertainty), I still don't feel clear and convincing enough regarding the explanation. Only one toy example doesn't make it clear on why it is more beneficial using only epistemic uncertainty, more studies with aleatoric uncertainty and without aleatoric uncertainty are needed to be conducted on the problems used in the paper to clearly demonstrate the benefit of using only epistemic uncertainty.
> >
> > The second concern is that I think it's important to discuss the design choices (hyperparameters) of the BNNs as they are important factors to ensure the accuracy of the surrogate model. If it is argued that any sets of hyperparameters will enable the similar levels of accuracy of the proposed method then this needs to be demonstrate clearly in the paper.
> >
> > Finally, I still maintain my opinion regarding the novelty of the proposed acquisition function, I think it is not novel enough by optimizing both the prediction and the uncertainty and use NSGA-II to optimize this acquisition function. For a simple method, I normally expect some deep analysis on how and why the method works well but the paper doesn't provide such analysis.
> >
> > For all these reasons, I still maintain my score as is.

---

> ### Author Response · Authors · 2023-11-23
>
> Thank you for your insightful feedback, we hope the answer below will address the raised concerns.
> ## More studies with and without aleatoric uncertainty using the problems in the paper.
> Given your feedback and the one of Reviewer 9JiQ, we conducted additional experiments to study the effect of including the aleatoric uncertainty in our acquisition function. These experiments are shown in the appended document (Rebuttal PDF_LBN_MOBO_aleatoric.pdf). We used the examples and procedures described in Sections C1.4 and C1.5 of the paper. The outcomes indicate that incorporating aleatoric uncertainty does not affect the exploration process in a beneficial manner—as showcased by the ZDT1 results—and may even delay the convergence, as observed in the ZDT2 and ZDT3 experiments. This is attributed to the nature of the aleatoric uncertainty, that measures inherent noise which cannot be diminished. This follows our original intuition that aleatoric uncertainty cannot be an effective tool for driving the exploration. If the reviewer sees any reason for further studies, we can repeat these experiments for our 3D printer and airfoil applications as well. Please instruct.
> ## More intuition on using only epistemic uncertainty and its benefits.
> **Why only epistemic uncertainty?**
> - Epistemic uncertainty is crucial for exploration as it identifies knowledge gaps. Aleatoric uncertainty, hinting at the intrinsic noise, doesn't aid exploration.
> - The training of prediction networks for epistemic uncertainty is fully independent from aleatoric uncertainty process. This ensures that ignoring aleatoric uncertainty introduces no errors in estimating epistemic uncertainty [1].
>
> **What are the benefits of using only epistemic uncertainty?**
> - Training with only epistemic uncertainty, yielding a faster and more stable training without the need to manage aleatoric uncertainty networks (variance networks) [1].
> - Omitting aleatoric uncertainty reduces model complexity, leading to fewer hyperparameters and a more straightforward training phase without balancing different uncertainties [2].
>
>
> ## Hyperparameter of the BNNs.
> The main focus of our paper is on the utilization of the simplified Deep Ensembles as the surrogate model. The main benefit of this surrogate is that it does not introduce any additional hyperparameters and it is sufficient to ensure that the sub-networks in the ensemble are trained accurately.
> Regarding other evaluated BNNs, we have adopted the implementation and the recommended hyperparameters from [3] and we also made sure that the training accuracy are reasonable. We will report the training accuracies for all methods in the final paper.
>
> ## Deeper analysis of the method.
> Please see an explicit list of our analyses in response to Reviewer s2Pz.
> In addition, we have observations to believe that the regret (the difference between the ideal and the current solution) grows at most sub-linearly. We will include in the revised paper (at least an empirical study) to show the regret growth in a batch scenario.
>
>
> [1] Nix, David A., and Andreas S. Weigend. "Estimating the mean and variance of the target probability distribution." Proceedings of 1994 ieee international conference on neural networks (ICNN'94). Vol. 1. IEEE, 1994.
>
> [2] Ansari, Navid, et al. "Autoinverse: Uncertainty aware inversion of neural networks." Advances in Neural Information Processing Systems 35 (2022): 8675-8686.
>
> [3] Li, Yucen Lily, Tim GJ Rudner, and Andrew Gordon Wilson. "A Study of Bayesian Neural Network Surrogates for Bayesian Optimization." arXiv preprint arXiv:2305.20028 (2023).

---

### Author Response · Authors · 2023-11-21
**Novelty and simplicity of the method.**

We thank the reviewers for their thoughtful feedback. We are delighted to receive positive feedback in general. We also acknowledge the constructive concerns of some reviewers which we believe can be fully addressed in the rebuttal (and later in the revised paper) and will help the paper improve considerably. We summarize here the main aspects of our rebuttal followed by our detailed response to each reviewer.
To reiterate our contributions: our paper introduces a new Bayesian optimization method (LBN-MOBO) that addresses an emerging class of problems in science and engineering where large-batches of data can be obtained during one round of experiments. The significance of this method is that it empowers scientists to embark on data-intensive explorations. In addition to obvious efficiency gains supported by a wide range of experiments, our method has deeply important implications. LBN-MOBO:
1. Shifts the bottleneck from the computational algorithm to our capacity for carrying out parallel experiments.
2. Features a minimal number of hyperparameters in both surrogation and acquisition stages. Also, the method is very easy to debug given the small number of modules involved.
3. Is very practical and can be implemented in less than a day.

To the best of our knowledge, and as evident from the reviews, both the strategy and its implications don't have any precedent in the literature.

Future research can:
1. Explore the possibility of making LBN-MOBO robust to aleatoric noise.
2. Investigate the possibility of incorporating complex constraints to be imposed on the candidates.
3. Can incorporate LBN-MOBO in various fields of studies to empower design and discovery in the scales that was not tractable before.

Given the valuable feedback by the reviewers, we have made specific changes and ran additional experiments:
1. We have provided a more challenging experiment in Section C3 of the revised paper in order to support the evidence for practicality of Deep Ensembles and dropout surrogate models (Section 5.1).
2. We have provided insightful information regarding the combination of 2MD acquisition function with an arbitrary surrogate model (Section B2 of the revised paper).
3. Inspired by reviewers' feedback, we have evaluated the possibility of using aleatoric uncertainty to improve the robustness of LBN-MOBO against noise. Since it is going to be a future work we include it as a separate supplementary material (Rebuttal PDF-LBN_MOBO_noisy.pdf).
4. Also, we have made several modifications in the main text of the paper to reflect the suggestions of the reviewers.

We have uploaded a revised version of the paper and the supplementary materials (SM). They will be further extended and polished for the final version. **Note that all new additions are highlighted in orange.**

---

### Meta-Review · Area_Chair_Utjd · 2023-12-05

**Metareview:**

This paper studies a Bayesian optimization setting with a large batch size where the objective is to reduce the required number of iterations instead of the number of samples. It formulates it as a multi-objective problem to optimize the mean and variance with an evolutionary algorithm. The design choices of the model and acquisition function are based on practicality with the very large batch size.

This paper receives divergent reviews on its quality. While multiple reviewers appreciate its simplicity and consider the problem under study to be important, there are some common concerns.

- Design choice on the use of Deep Ensemble as the model; NSGA-II as the acquisition optimizer; mean, variance multi-objective vs UCB

Authors' detailed response should have addressed this question

- Limited novelty: The proposed solution is to treat the problem as a multi-objective optimization and use established components.

While some reviewer appreciates its simplicity, other reviewers consider it as a weakness.

- Mainly studies noiseless setting and only consider modeling the epistemic uncertainty

This is the main remaining concern after rebuttal. The extension to noisy environments is important for broader real-world applications but requires a considerable amount of analysis on epistemic vs aleatoric uncertainty, additional experiments. While the authors provide additional experiments and discussion in the rebuttal, the amount of required changes to the current version would deserve another round of reviewing.

Based on the remaining concerns and the amount of analysis and experiments requested in the rebuttal period, the current submission is below the threshold of acceptance. I would strongly encourage the authors to incorporate the comments and new results in the discussion and prepare a new revision for a future submission.

**Justification For Why Not Higher Score:**

Divergent options after discussion. Concerns are shared among multiple reviewers.

**Justification For Why Not Lower Score:**

N/A

---

### Decision · Program_Chairs · 2024-01-16

Reject